# Widespread bacterial diversity within the bacteriome of fungi

Aaron J. Robinson [1], Geoffrey L. House[1], Demosthenes P. Morales[1,2], Julia M. Kelliher [1], La Verne Gallegos-Graves[1], Erick S. LeBrun [1], Karen W. Davenport[1], Fabio Palmieri [3], Andrea Lohberger[3], Danaé Bregnard [3], Aislinn Estoppey[3], Matteo Buffi[3], Christophe Paul [3], Thomas Junier[3], Vincent Hervé [3], Guillaume Cailleau [3], Simone Lupini[4], Hang N. Nguyen [4], Amy O. Zheng[5], Luciana Jandelli Gimenes[6], Saskia Bindschedller[3], Debora F. Rodrigues[4], James H. Werner[2], Jamey D. Young [5], Pilar Junier [3] & Patrick S. G. Chain [1✉]

Knowledge of associations between fungal hosts and their bacterial associates has steadily grown in recent years as the number and diversity of examinations have increased, but current knowledge is predominantly limited to a small number of fungal taxa and bacterial partners. Here, we screened for potential bacterial associates in over 700 phylogenetically diverse fungal isolates, representing 366 genera, or a tenfold increase compared with previously examined fungal genera, including isolates from several previously unexplored phyla. Both a 16 S rDNA-based exploration of fungal isolates from four distinct culture collections spanning North America, South America and Europe, and a bioinformatic screen for bacterial-specific sequences within fungal genome sequencing projects, revealed that a surprisingly diverse array of bacterial associates are frequently found in otherwise axenic fungal cultures. We demonstrate that bacterial associations with diverse fungal hosts appear to be the rule, rather than the exception, and deserve increased consideration in microbiome studies and in examinations of microbial interactions.

[1] Biosecurity and Public Health Group, Bioscience Division, Los Alamos National Laboratory, Los Alamos, NM 87545, USA. [2] Center of Integrated Nanotechnologies, Los Alamos National Laboratory, Los Alamos, NM 87545, USA. [3] Laboratory of Microbiology, Institute of Biology, University of Neuchâtel, CH-2000 Neuchâtel, Switzerland. [4] Department of Civil and Environmental Engineering, University of Houston, Houston, TX 77004, USA. [5] Department of Chemical and Biomolecular Engineering and Department of Molecular Physiology and Biophysics, Vanderbilt University, Nashville, TN 37235-1604, USA. [6] Center for Environmental Research and Training, University of São Paulo, Cubatão, São Paulo 11.540 -990, Brazil. ✉email: pchain@lanl.gov

Microbiome research permeates all domains of biology and the concept of the holobiont (both host and the host-associated microbial community treated as a combined unit) is changing the perception of individual biological eukaryotic units[1]. Fungi have been primarily considered as one of the microbial components of the host-associated microbiome in numerous animal and plant studies[2–4] and are often overlooked in human studies[5]. However, the existence of a fungal bacteriome, consisting of bacteria found both within and in close association with cells of a fungal host, is an emerging concept[6].

The field studying bacterial–fungal interactions (BFI) is complex and dynamic, with research ranging from specific bacterial–fungal interactions to larger scale community analyses[7]. While community-level studies of environmental co-occurrence between bacteria and fungi are important and have revealed interesting patterns[8–10], such community-level correlations do not reveal specific fungal-bacterial associations. For example, some bacteria and fungi that co-occur under similar environmental conditions, but never actually interact with each other, could be highlighted in a co-occurrence study, but likely be excluded when examined for BFI specifically. Examining bacterial associates of fungal hosts that have been isolated and maintained in otherwise axenic culture removes any uncertainty about the fungal partner, while providing greater confidence that the bacteria form a tight and long-term association with the fungal host. Focusing on fungal isolates has the aforementioned benefits, but distinguishing between true biological associations and other factors resulting in the detection of a bacterial signature (e.g. co-isolation or contamination) can still be difficult without detailed and time-intensive investigations of each putative association. The current catalog of known associations between bacteria and fungal isolates is largely the result of specific examinations of endohyphal bacterial associates of mycorrhizae and plant-associated fungal endophytes, and is focused primarily on a small number of fungal genera from either the early diverging fungal phylum Mucoromycota[11–13] or the large and highly diverse subkingdom Dikarya (Ascomycota and Basidiomycota)[14–17]. Despite the growing number of examples of specific bacterial associations with isolated fungal hosts, a broader perspective of both intra and extracellular bacterial associates among the larger diversity of fungal hosts is necessary in order to understand the potential evolutionary and ecological consequences of these interkingdom interactions.

To address this knowledge gap and gain a more comprehensive view of the diversity of bacterial–fungal associations, we employed two complementary approaches to identify signals of potential bacterial associates among a phylogenetically broad range of fungi. We analyzed 16S ribosomal RNA (rRNA) gene amplicon sequences obtained from total DNA extractions of distinct fungal isolates belonging to four different fungal culture collections from diverse environments within North America, South America, and Europe (hereafter referred to as 16S-CC screen). We also searched for bacterial-specific sequences (hereafter referred to as BSS screen) within publicly available fungal genome sequencing projects from the Joint Genome Institute (JGI) Mycocosm portal,[18] deliberately sampling the widest possible range of fungal phylogenetic diversity. Stringent quality control standards and procedures were employed for both screens, to increase confidence in the accuracy of the observed results. This work provides a considerably more comprehensive exploration of the fungal bacteriome by examining over 700 fungal isolates, including 366 fungal genera (nearly ten times the amount in all previous examinations) and multiple representatives from six out of the eight recognized fungal phyla. Putative bacterial associations were found to be both common and complex, with an unexpected diversity of bacteria detected across all

examined fungal lineages. This raises a multitude of questions: how frequently do fungi serve as hosts for specialized bacteria, or as potential substrates for bacteria with broad ecological niches? To what extent are bacteria transient or more persistent in their occurrence with fungi? Considering both transient co-occurrences and more persistent associations, what are the potential roles and impacts of the bacteriome on fungi, including impacts on interactions with other microscopic (e.g. protists) or macroscopic (e.g. plants and animals) organisms?

## Results

Two complementary methods were used to explore the diversity of constituents of the fungal bacteriome: an amplicon community profiling survey of four independent and predominantly soil-derived culture collections from three geographical origins (Europe, South America and North America), and a scan of fungal genome sequencing projects for bacterial genomic signatures.

**Fungal diversity in four culture collections**. The internal transcribed spacer (ITS) rDNA region was used to characterize the diversity of 294 cultivable fungal isolates from four culture collections (Supplementary Data 1). ITS amplicons obtained via Sanger sequencing were used to taxonomically classify each examined fungal isolate based on strict identity thresholds using the UNITE and BLAST databases (see Methods section). These isolates, which were obtained from Europe, North America, and South America, represent 4 phyla, 15 classes, 32 orders, 67 families, and 93 genera (Supplementary Data 2). Members of the Ascomycota (67%) and the Basidiomycota (25%) dominated the collections. A total of 86 genera from these two phyla (49 Ascomycota and 37 Basidiomycota) were examined in our 16S-CC screen, including 7 genera with previously reported bacterial associates (Fig. 1a and Supplementary Data 3). Fungal isolates from the phylum Mucoromycota (8%) and Zoopagomycota (0.7%), which are commonly found in symbiotic associations with plants (Mucoromycota) and animals (Zoopagomycota)[19], were less common in our collections. Our 16S-CC screen included five genera from the Mucoromycota, all of which have previously described bacterial associates (*Mortierella, Mucor, Podila, Rhizopus, Umbelopsis*)[20–24]. This study also represents the first examination of bacterial associates in the Zoopagomycota (Fig. 2).

**Screening fungal isolates reveals remarkable bacterial diversity in all fungi**. Every fungal isolate examined in the 16S-CC screen harbored at least one putative bacterial associate (Supplementary Data 4). Across all examined fungal isolates, a total of 6594 16S amplicon sequencing variants (ASVs) were clustered into 705 bacterial operational taxonomic units (OTUs), based on ASV taxonomic classification (see Methods section), representing 27 phyla, 53 classes, 108 orders, 213 families, and 546 genera. A total of 134/6594 of these bacterial ASVs (~2%), representing 49 bacterial genera, were found in fungal isolates from two (102) or three (32) of the examined collections. Only 73 (~13%) of the 546 bacterial genera detected in this screen were previously described as possible associates of fungi (Fig. 1b). When compared with the diversity of bacterial associates from all previous studies combined, this result represents a substantial expansion at all taxonomic levels, including 12 new phyla and 471 new genera (Fig. 2 and Supplementary Data 5).

Interestingly, we found a wide range of bacterial richness per fungal isolate (1–100 OTUs), with an average of 34 OTUs per isolate, indicating that it was typical for diverse bacteria to co-exist within the examined fungal isolates, many of which have been propagated for years (Supplementary Fig. 1a). There appear

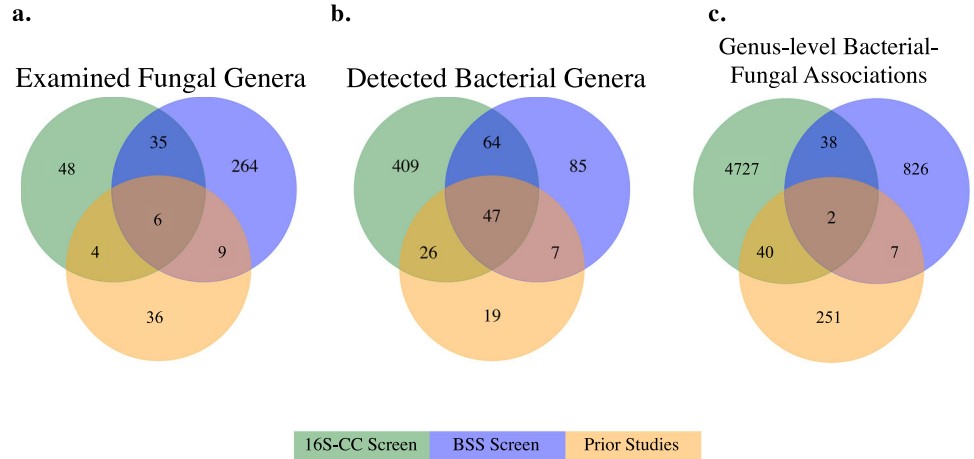

**Fig. 1 Fungal taxa, bacterial taxa, and genus-level associations found among the culture collection screen (16S-CC), the bioinformatic screen of fungal genome sequencing projects (BSS) and prior studies.** Each diagram displays overlaps in either **a** examined fungal genera, **b** bacterial genera detected, or **c** genus-level bacterial–fungal associations found in the two screens conducted as part of this work and in prior studies.

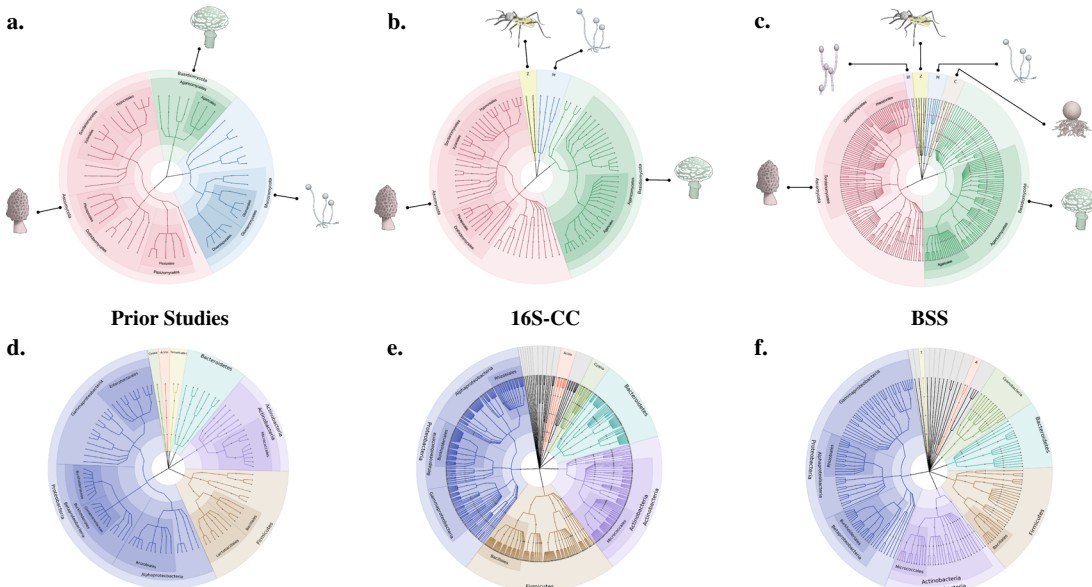

**Fig. 2 Expanded taxonomic diversity of bacterial–fungal associations.** Prior findings of bacterial–fungal associations were compared with both the 16S-CC and BSS screens completed in this study. The upper panel presents the taxonomic diversity of fungal hosts examined in **a** prior studies, **b** the culture collections included in the 16S-CC screen, and **c** the fungal genome projects used in the BSS screen. The lower panel displays the corresponding taxonomic diversity of bacteria associated with these fungi, observed in **d** prior studies, **e** the 16S-CC screen, and **f** the BSS screen. Terminal nodes represent distinct genera and edges are colored by phyla. Abbreviations for the fungal hosts: M Mucoromycota, Z Zoopagomycota, Bl Blastocladiomycota, C Chytridiomycota. Abbreviations for the bacterial associates: Acido/A Acidobacteria, Cyano Cyanobacteria, T Tenericutes). Several bacterial phyla not previously described as fungal associates, but identified in our 16S-CC and the BSS screens are represented in gray (Aquificae, Armatimonadetes, Calditrichaeota, Chlamydiae, Chloroflexi, Deinococcus-Thermus, Fibrobacteres, Fusobacteria, Gemmatimonadetes, Nitrospirae, Planctomycetes, Rhodothermaeota, Spirochaetes, Synergistetes, Thermodesulfobacteria, Thermotogae, and Verrucomicrobia).

to be no correlations between taxonomy of the fungal host and the number of putative bacterial associates, as all examined fungal phyla had a few isolates with above average numbers of bacterial associates. A total of 464 bacterial OTUs were detected in five or fewer fungal isolates, with 182 bacterial genera found to each occur in only a single fungal isolate (Supplementary Fig. 2). This pattern suggests that specific, or possibly opportunistic, interactions are also not uncommon. These 182 isolate-specific bacterial genera occurred within 96 fungal isolates, indicating that some fungal isolates harbored more than one specific bacterial OTU. This observation was found among all four fungal phyla investigated. Comparisons with previous work revealed that only

seven of these 182 isolate-specific bacterial genera (*Chitinophaga, Cohnella, Erwinia, Lachnoclostridium, Moraxella, Rhodopseudomonas,* and *Sphingobium*) were previously described associates of fungi, but in all cases, they associated with different fungal genera than those observed in this screen (Supplementary Data 3)[15–17,25–27].

Members of the bacterial lineages Betaproteobacteria (found in 271 of 294 fungal isolates), Gammaproteobacteria (258), Alphaproteobacteria (256), Actinobacteria (247), and Bacilli (219) were detected most frequently across all examined fungal isolates and OTUs assigned to these five lineages were also responsible for a large (6411; ~82%) proportion of all 7830 detected bacterial–fungal

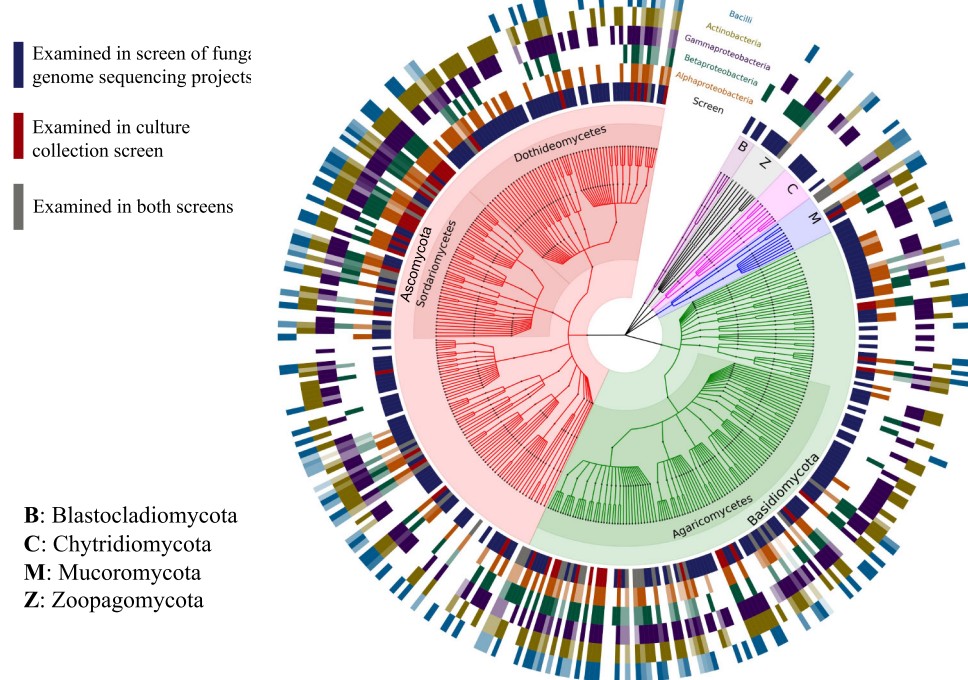

**Fig. 3 Diversity of bacteria found among fungi from culture collections and public fungal genome sequencing projects.** The internal cladogram is colored by fungal phyla and terminal nodes represent fungal genera. The innermost ring indicates the screen used to detect these bacterial–fungal associations, and the outer five rings represent the top five bacterial lineages (from inner to outer ring: Alphaproteobacteria, Betaproteobacteria, Gammaproteobacteria, Actinobacteria, and Bacilli) most frequently detected in our screens. These rings are shaded from dark (many interactions detected) to white (no interactions observed).

associations (Fig. 3). *Corynebacterium* (found in 146 of the 294 fungal isolates; Actinobacteria), *Massilia* (143; Betaproteobacteria), *Streptococcus* (142; Bacilli), *Brevundimonas* (142; Alphaproteobacteria), and *Sphingomonas* (129; Alphaproteobacteria) were the most frequently detected bacterial OTUs and were also found in isolates from all four examined fungal phyla. These bacterial genera have been found to be common soil inhabitants and, with the exception of *Brevundimonas*, these common bacterial genera from the 16S-CC screen were also previously described as associating with fungi[14,15,17,25,28,29]. While 38 previously described bacterial–fungal associations were corroborated using this screen, we report a more than 100-fold increase with an additional 4818 putative genus-level bacterial–fungal associations (Supplementary Data 3).

Our 16S-CC screen, which greatly expands our knowledge of the diversity of putative bacterial associates, has revealed a number of novel potential bacterial partners even in well-examined fungal systems, such as within the genus *Mortierella* (Mucoromycota)[30–32]. We detected 89 bacterial genera not previously described as associates of *Mortierella*, although 27 of these have been previously found as associates of other fungi (Supplementary Data 3). Comparisons of the novel bacterial associates of these *Mortierella* isolates revealed that 51 of the 89 bacterial genera were found in more than one *Mortierella* isolate (Supplementary Fig. 3). The most common bacterial genera found associating with these *Mortierella* isolates were *Arthrobacter*, *Corynebacterium*, *Delftia*, *Micrococcus*, and *Streptococcus*, which were also common associates of other fungal genera examined in this screen (Supplementary Data 3). Interestingly, the three *Mortierella* isolates lacking *Arthrobacter*, *Corynebacterium*, and *Streptococcus* appear to have a separate 'core'

bacteriome consisting of *Bradyrhizobium*, *Dyella*, *Hydrotalea*, *Mesorhizobium*, and *Terrimonas* (Supplementary Fig. 3). While none of these bacterial genera were previously described associates of *Mortierella*, previous work suggests the co-occurrence of *Terrimonas* with *Mortierella* in soils[33]. Our screen also included an isolate of *Podila*, a genus which until a recent reclassification, was considered a lineage of *Mortierella*[34]. Consistent with its only recent reclassification, it appears that the single *Podila* isolate had a similar bacteriome profile to some of the other examined *Mortierella* isolates (Supplementary Fig. 3).

Despite this finding of potential 'core' bacteria found among multiple *Mortierella* isolates, each isolate had its own distinct bacteriome. Outside of these two potential 'core' bacteriome taxa, we found a number of bacterial taxa that were found only in one *Mortierella* isolate, and another group of bacterial taxa that were found in two or more isolates, but that did not share co-occurrence patterns that could easily be distinguished (Supplementary Fig. 3). Thus, while the bacteriomes of closely related *Mortierella* fungi can share a number of bacterial taxa, both with other *Mortierella* isolates and close relatives such as the examined *Podila* isolate, each isolate's bacteriome can also maintain aspects of uniqueness.

**Validation of bacterial–fungal associations using microscopy.**
To complement the results of the 16S-CC screen and further explore some of the detected bacterial associations, we utilized fluorescence in situ hybridization (FISH) techniques targeting the 16S rRNA to visualize the presence and localization of bacterial associates for seven diverse fungal isolates examined in the 16S-CC screen. Given the large number of fungal isolates examined

overall, it would have been impractical to attempt FISH staining on every single isolate. Instead, fungal isolates were selected due to their filamentous growth characteristics (fast linear mycelial growth with minimal strand overlap), which are ideal for imaging, and because these isolates represent fungal genera with the highest number of isolates in the 16S-CC screen (*Alternaria*, *Aspergillus*, *Fusarim*, *Ilyonectria*, *Penicillium*, *Rhizopus*, and *Trichoderma*). To our knowledge, bacterial associates have not been previously reported for isolates of *Ilyonectria*, increasing the need to further validate their associations with bacteria. Bacterial cells were visible in all seven examined fungal cultures (Fig. 4, wider fields of view of fungal samples provided in Supplementary Fig. 4). We anticipated that the local concentration of bacteria in hyphae and ribosomal targets would vary among fungal isolates, therefore, we tested two in situ staining techniques to acquire the most pronounced signal relative to background (Supplementary Fig. 5). We first employed a direct hybrid probe fluorescently labeled on the 5′ and 3′ ends, and if the signal was too low, we used a fluorescence amplification technique known as hybridization chain reaction (HCR) FISH[35,36]. Distinct bacterial morphologies were difficult to resolve in all cases, but given the high bacterial diversity reported in the 16S-CC screen for these fungal isolates and our use of conserved 16S probes, this was anticipated. Several interesting observations were made from these experiments, for instance, the bacterial staining results for the *Rhizopus* isolate from our 16S-CC screen resembled previously published FISH experiments of other *Rhizopus* isolates with described bacterial associates[37]. In the *Aspergillus* isolate, high density clusters of bacteria resembling biofilms were observed. These microscopy examinations confirm the presence of bacterial associates among these diverse fungal isolates and demonstrate the spatial patterns and variability of bacterial associates among different fungal isolates and even within the hyphae of a single fungal host, such as is observed in adjacent hyphae not exhibiting bacterial signals (indicated by stars in Fig. 4).

To demonstrate the utility of the 16S ASV data to generate unique FISH probes specific to certain bacterial taxonomic groups, we selected an *Aspergillus* isolate from one of the culture collections (LANL.1351.96) with bacterial ASVs representing 32 distinct genera (Supplementary Data 6). This isolate was selected due to both the diversity of bacterial ASVs found, and our previous success imaging *Aspergillus* isolates (Fig. 4). To ensure the specificity of these probes, in silico off-target analysis was performed comparing the computed FISH probe sequences against all available bacterial genomes, including bacterial groups not represented in the ASVs identified in this isolate, as well as to multiple *Aspergillus* genomes (see Methods section). The bacterial genus *Lacunisphaera* was selected as one of our targets to validate our findings with a novel putative associate, as members of this bacterial genus have not previously been described as associates of any fungi, and the relative abundance of *Lacunisphaera* ASVs detected in this *Aspergillus* isolate were quite low (31 and 60, from a total of 9922 sequences). A target with a lower relative abundance was selected to aid in gauging the sensitivity of the probes, and to increase confidence that these potential bacterial associates are yet present, despite their low relative abundance.

Our method produced a pool of 11 FISH probe sequences specific to this bacterial genus, to increase the likelihood of detection even at relatively low abundance relative to other bacterial associates. Visualization of the *Lacunisphaera* 16S rRNA in the *Aspergillus* isolate using fluorescence microscopy is shown in Fig. 5 and depicts morphologies similar to previous reports of *Lacunisphaera* shape and size[38], together with mono- and diplococci-like structures (white arrows). DNA staining using DAPI also displayed similar structures correlating with the 16S

signal, which were distinct from the *Aspergillus* nuclei indicated by the white asterisks. 3D projections of the figure are provided in Supplementary Movie 1 to demonstrate the localization of the bacterial signal within the fungal hyphae.

**Fungal genome sequencing data reveal unexpected bacterial associates in previously unexplored fungal lineages.** While amplicon studies of existing culture collections help to uncover the breadth of potential bacterial diversity associated with fungal cultures, we further expanded the diversity of examined fungal lineages by investigating 408 fungal genome sequencing projects (314 fungal genera) from the JGI Mycocosm portal[18] (Supplementary Data 7), which harbors the most diverse collection of fungal genome sequencing projects to date. The diversity of fungal hosts examined for bacterial-specific sequences (BSS) is extensive and expanded our examination by adding 264 fungal genera not previously considered, including nine genera from two new phyla (Blastocladiomycota and Chytridiomycota) that have never been previously examined for bacterial associates, as well as substantial expansions within the Ascomycota and Basidiomycota (Fig. 2). The remaining 50 fungal genera in this dataset were either previously described in other studies as being associated with bacteria (9 genera), determined to harbor bacteria using our 16S-CC screen above (35), or both (6) (Fig. 1a).

Under the hypothesis that sequences of bacterial associates would only constitute a very small fraction of the genomic data, only projects sequenced using high-throughput (e.g. Illumina and 454) technologies were examined to maximize the probability of capturing BSS. Analysis performed with the sensitive and highly specific classification algorithm GOTTCHA2[39], revealed that bacterial signatures were detected in the majority (323/408, or 79%) of the examined fungal genome projects (Supplementary Data 8 and Supplementary Data 9).

While this BSS screen focuses specifically on the unique genomic fraction of any given bacterial genus (unique signatures), we found that on average, >55% of the unique signatures per genus were recovered, representing an average of over 241.5 kb of bacterial genomic signal at an average depth of coverage of 54 fold. The breadth and depth of unique (genus-specific) genome coverage that was captured for any given bacterial genus suggests a substantial relative number of bacterial cells associated with the fungal host, which also implies an active functional relationship with the fungal host. A graphical representation of the number and percentage of bacterial reads identified in each fungal genome sequencing project, as well as the proportion of each bacterial phylum detected in each project are presented in Supplementary Fig. 6.

Genomic signatures of putative bacterial associates were detected in representatives from every examined fungal phylum. The average number (3) and range (1–20) of different bacterial genera identified per fungal isolate were substantially lower than what was observed in the 16S-CC screen (Supplementary Fig. 1). This result was expected given that the sequencing projects were focused on the fungal genome and that any bacterial associates would constitute a considerably smaller proportion of the sequencing output compared with the host genome. This is in clear contrast with the 16S-CC screen, which is based on targeted amplification of bacterial rRNA, increasing the likelihood of identifying bacterial associates present at low levels.

In terms of the taxonomic diversity of bacterial genera found associated with fungi, a smaller number of genera were discovered using the BSS screen relative to the 16S-CC screen, but the overall taxonomic distribution was proportionally similar (Fig. 2). The most frequently observed bacterial classes were the same between this BSS screen and our 16S-CC screen, with *Escherichia* (105 of the 408 fungal isolates), *Stenotrophomonas* (72), *Cutibacterium*

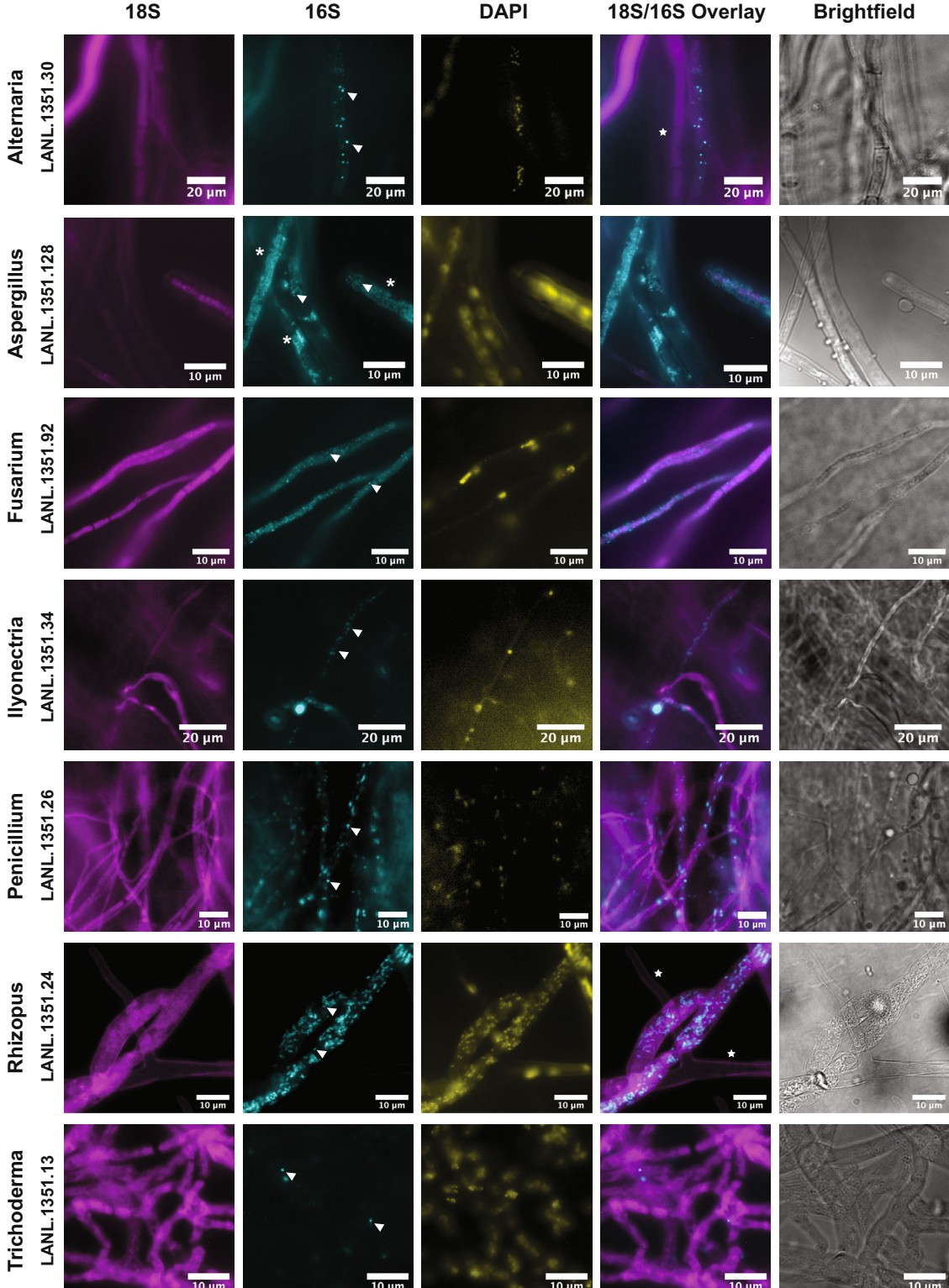

**Fig. 4 Visualization of bacterial associates in diverse fungal hosts using 16S ribosomal RNA staining by fluorescence in situ hybridization.** Fungal ribosomes were stained using a universal eukaryotic 18S rRNA probe (magenta); bacteria were co-stained with a universal 16S rRNA probe (cyan); DAPI was used as a global nuclear stain (yellow). Overlays of the 18S and 16S fluorescence show positive correlation of bacteria along or within hyphae. Observed bacterial signal was variable among fungal samples, and displayed coccoid and rod-shaped phenotypes. Individual structures resembling bacterial cells were observed (arrowheads) as were biofilm-like growth patterns along hyphae (asterisks). Bacterial structures absent in neighboring hyphae indicate variability in spatial distributions (stars).

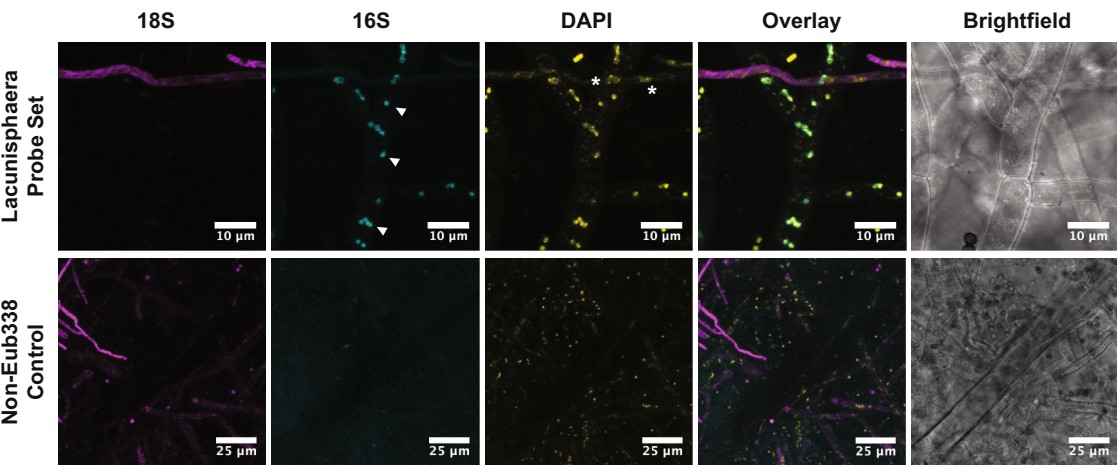

**Fig. 5 Fluorescence in situ hybridization staining with *Lacunisphaera* specific probes in an *Aspergillus* isolate.** FISH staining was conducted on an *Aspergillus* isolate using a universal 18S rRNA probe (magenta) and a *Lacunisphaera* genus-specific 16S rRNA probe set (cyan). Coccoid structures attributed to the bacterial associate are shown with arrowheads. Non-specific DAPI DNA staining displays co-localization with 16S signals from the bacteria that appear distinct from the nuclear morphology of the fungus (asterisks). A non-hybridizing probe (Non-Eub338) was used as a negative control.

(72), *Rhodococcus* (45), and *Methylobacterium* (35) as the most frequently detected genera in this screen. Four of these most frequently detected genera have been previously described as fungal associates (*Escherichia*, *Methylobacterium*, *Stenotrophomonas*, and *Rhodococcus*)[15,21,26–28], but in different fungal genera than those observed here. These results further support the notion that some bacterial groups form diverse and unspecific associations with multiple fungal hosts.

In total, more than half (111 out of 203) of the bacterial genera detected in this BSS screen had also been identified in the 16S-CC screen, while 85 genera were detected only in this screen (Fig. 1b). These 85 bacterial genera were found among 76 fungal genera, 63 of which were examined exclusively in this BSS screen, indicating these novel bacterial associates can be largely explained by the inclusion of previously unexamined fungal genera. Several of the remaining genus-level associations detected in this BSS screen were also present in the 16S-CC screen (38 shared associations), other previous examinations (7 shared associations), or found both in the 16S-CC screen and previous examinations (2 shared associations; Fig. 1c).

**Complementary methods converge on bacterial–fungal associations.** The frequency and diversity of putative bacterial associations detected in the two complementary large-scale screens resulted in a complex network of possible bacterial–fungal associations that make overarching statements and interpretations challenging (Fig. 3 and Supplementary Data 4). Phylum-level comparisons of the observed bacterial–fungal associations from both the BSS and 16S-CC screens indicated potential patterns of association between certain bacterial and fungal taxa (Fig. 6a). However, no clear patterns of association were apparent at the genus-level (Fig. 6b). The genus-level comparison demonstrates the complexity of the bacterial associations detected in both screens, and indicates the patterns observed in the phylum-level comparisons are not well supported at other taxonomic scales. Because of the bias in sampled fungal isolates, with some genera and classes represented more frequently than others, we focused our efforts on analyzing the core components of the complex network of interactions that were supported by both the 16S-CC and BSS screens. Several genera from the Ascomycota and Basidiomycota, as well as two genera from the Zoopagomycota are highlighted using this approach (Fig. 6c). Only four (*Morchella*, *Penicillium*, *Suillus*, and *Trichoderma*) of the 17 fungal genera detected using this network

analysis had previous descriptions of bacterial associates[15,26,27,40], while 13 of the 19 bacterial genera (*Bacillus*, *Burkholderia*, *Enterobacter*, *Paraburkholderia*, *Rhodococcus*, *Staphylococcus*, *Stenotrophomonas*, *Corynebacterium*, *Comamonas*, *Acinetobacter*, *Lactobacillus*, *Pseudomonas*, and *Rhizobium*) were known associates of fungal isolates[14–17,21,25–29]. Many of these associations were also detected in several fungal isolates, in either or both of our screens. Independent observations of specific associations between the same fungal and bacterial genera, particularly when utilizing diverse screening methods and isolates, suggests that these associations could be beneficial or essential for one or both participants.

While several distinct associations were apparent in the high-level network analysis, we further examined finer scale patterns of taxonomic associations. We selected our most sampled fungal genus (*Aspergillus*) to examine in detail whether closely related isolates shared similar profiles of bacterial associations. In an attempt to minimize method specific variance, this analysis was limited to bacterial partners of *Aspergillus* that were detected in both screens. Our examination of 51 *Aspergillus* isolates, including 23 described species from the BSS screen, revealed that while some general patterns of association were present, closely related *Aspergillus* isolates can have distinct profiles of bacterial associates (Supplementary Fig. 7). These comparisons are possible in *Aspergillus* given the magnitude and evenness of isolate sampling across both screens, but many of the fungal genera examined were only represented by a few or a single isolate, making similar comparisons for different fungal taxa challenging or impossible. Regardless, this demonstrates that even closely related fungi can greatly vary in their bacteriome communities, providing further support for the observed complex network of bacterial–fungal associations.

**Similarity and phylogenetic diversity among bacterial associates of fungi.** Unlike the 16S-CC screen, where sequencing data was limited to a fragment of the 16S region, the fungal genome sequencing projects examined in the BSS screen often captured much larger fractions of bacterial genomes, which in some cases, allowed for more comprehensive genome-wide evolutionary comparisons. We selected three bacterial species (*Acinetobacter johnsonii*, *Bacillus cereus*, and *Escherichia coli*) that were detected at high levels (see Methods section) by GOTTCHA2 in four or more taxonomically diverse fungal datasets to evaluate phylogenetic relationships between fungal associates of the same species

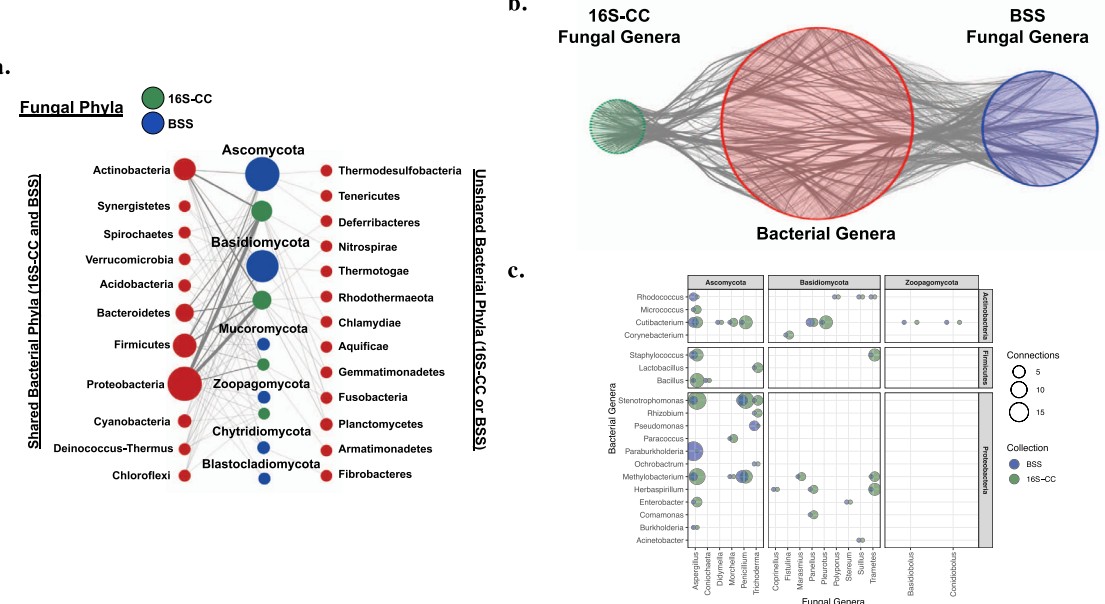

**Fig. 6 Phylum and genus level bacterial–fungal associations identified in fungal culture collections and genome sequencing projects. a** Tripartite network showing associations (gray edges) between bacterial phyla (red) and fungal phyla from either the 16S-CC (green) or BSS (blue) screen. Bacterial phyla that were identified in both screens are shown on the left, while phyla only identified in one screen are shown on the right. Nodes are sized by the number of genera represented in each phylum and edges are weighted by the number of distinct associations among bacterial and fungal genera. **b** Network of associations (gray edges) between bacterial genera (red nodes) and fungal genera from either the 16S-CC screen (green nodes) or the BSS screen (blue nodes) generated using Cytoscape, which demonstrates the complexity of the overall network of potential associations detected across both screens at the genus level. **c** Genus-level bacterial–fungal associations identified in both the 16S-CC and BSS screen. The size of each circle corresponds to the total number of genera detected in each screen.

We examined *B. cereus* sequences derived from the genome sequencing projects of four diverse fungal species: *Loramyces macrosporus* (Ascomycota, Leotiomycetes, Helotiales), *Melanomma pulvis-pyrius* (Ascomycota, Dothideomycetes, Pleosporales), *Mrakia frigida* (Basidiomycota, Tremellomycetes, Cystofilobasidiales), and *Phycomyces blakesleeanus* (Mucoromycota, Mucoromycetes, Mucorales). Whole genome alignments and phylogenetic analyses of these fungal dataset-derived sequences (i.e. a core genome size of 329,255 base pairs with 37,600 total SNPs) indicated that three of the *B. cereus* strains putatively associating with these fungi are closely related via a recent common ancestor (with average of 1136 SNPs in the core genome), while one strain associating with *M. pulvis-pyrius* appears to be more distantly related (17,704 SNPs compared with the other fungal-derived genomes) (Fig. 7a). Interestingly, the three most closely related *B. cereus* strains examined in this analysis were associates of fungal genera belonging to three separate phyla (Ascomycota, Basidiomycota, and Mucoromycota), while the *B. cereus* strains from the two most closely related fungal hosts, *L. macrosporus* and *M. pulvis-pyrius*, appear more distantly related. These results indicate that diverse fungi may harbor phylogenetically related bacteria, and that multiple lineages within a bacterial species may be able to form such cross-kingdom associations.

A phylogenetically less diverse series of potential associations were found using the same phylogenetic approach with *E. coli* and *A. johnsonii* sequences found in different fungal projects (Fig. 7b, c). These examinations utilized core genomes of *E. coli* (3,518,747 bp in length) and *A. johnsonii* (20,697 bp in length). A total of 84,954 (*E. coli*) and 930 (*A. johnsonii*) SNPs were utilized to infer phylogenetic trees. Fungal-derived strains from these two bacterial species appear to be closely related, with an average of only 5197 (*E. coli*) and 17 (*A. johnsonii*) SNPs detected within the

core genome across the analyzed strains. While the hosts of both bacterial taxa are phylogenetically diverse, the four examined fungal hosts of *A. johnsonii*, occupy similar ecological niches (despite representing two fungal phyla and four fungal orders), while the examined six hosts of *E. coli* lack this similarity. Two of these *A. johnsonii* fungal hosts are well characterized pathogens of woody plants (*Phaeoacremonium aleophilum* and *Grosmannia clavigera*) and the other two are white-rot saprotrophs of woody plants (*Panellus stipticus* and *Scytinostroma* sp.). These overlaps in the niche and trophic mode of the fungal hosts could be one possible explanation for the phylogenetic similarity found between their potential bacterial associates.

**Discussion**

The results presented in this study demonstrate for the first time that multiple potential bacterial associates are common in a large diversity of fungal isolates across all examined phyla and suggest the fungal bacteriome can be quite complex. Potential bacterial associates were detected in the vast majority (617 or 88%) of the 702 fungal isolates examined across both the 16S-CC and BSS screens, including representatives from three phyla previously unexamined for the presence of bacterial associates (Blastocladiomycota, Chytridiomycota, and Zoopagomycota). Quality control standards and procedures were employed throughout our investigations to increase confidence in the accuracy of the presented results and aid in interpretation. We consistently describe bacteria detected in our screens as 'potential' bacterial associates given that the nature of each interaction is not yet explored in depth. However, it is important to consider that bacteria capable of persisting alongside a presumably axenic fungal isolate in culture, without obvious signs of parasitism, regardless of their origin, should be considered at minimum a potential associate.

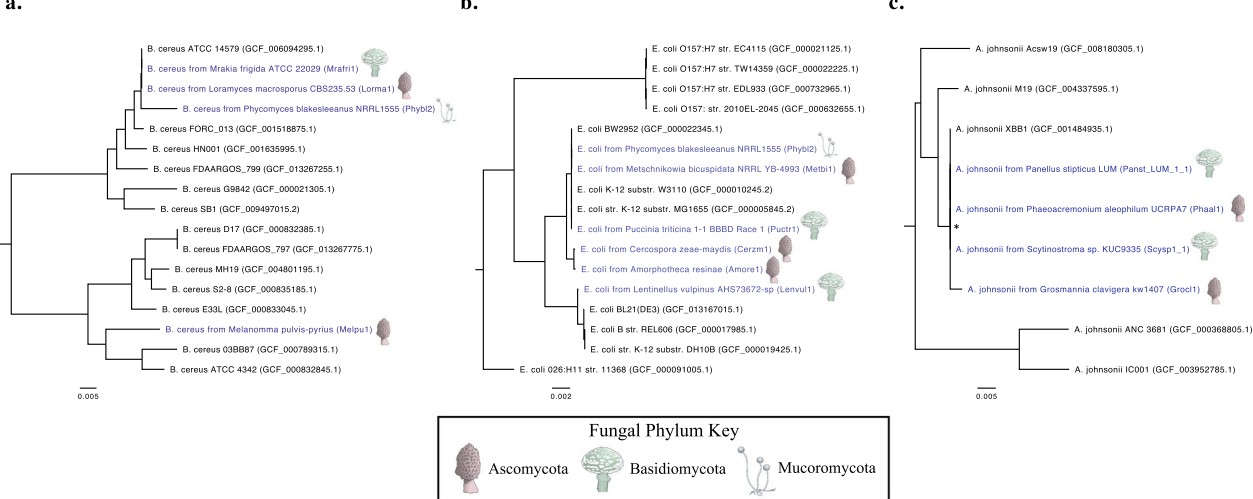

**Fig. 7 Genome-wide single nucleotide polymorphism (SNP)-based phylogenetic analysis of fungal-derived bacterial sequences.** Phylogenetic trees generated by analysis with PhaME for fungal-derived sequences of **a** *Bacillus cereus*, **b** *Escherichia coli*, and **c** *Acinetobacter johnsonii*. Fungal-derived sequences are annotated in blue lettering and include the corresponding JGI project ID (shown in parentheses) and icons next to their annotation indicate the phylum of the fungal host. Non-fungal-derived reference genomes are indicated in black. NCBI accession IDs are included for these reference genomes (shown in parenthesis). Branches with bootstrap values (1000 bootstrap replicates) <60% are marked with an asterisk (*). All trees are midpoint rooted.

Multiple representatives from both early diverging, and higher lineages of fungi were examined in this work, suggesting bacterial associations proliferate all branches of fungal evolution. The presence of potential bacterial associates in all examined lineages of early diverging fungi also suggests associations with bacteria arose early in the evolutionary history of fungi, which may be expected given that members of these kingdoms co-dominate most terrestrial environments and frequently overlap in environmental niches[41].

Our results indicate a remarkable, and previously undescribed, diversity of bacterial lineages that appear capable of associating closely with fungi. In our 16S amplicon screen of diverse fungi belonging to multiple culture collections, at least a single putative bacterial associate was detected in every examined isolate. This detection rate was unexpected given that results from previously published surveys usually contain several isolates lacking any potential bacterial associates. However, differences in the applied methodology of these previous surveys such as the use of primers targeting specific taxonomic groups or the treatment of fungal tissue to eliminate or reduce external bacterial associates, compared with the more sensitive methods applied here, could explain the increased discovery rate, as our methods are designed to generally detect any potential bacterial associates.

A total of 638 bacterial genera were detected across both screens, including representatives from 17 bacterial phyla that have not previously been described as having associations with fungi. Members of the Proteobacteria, Actinobacteria and Firmicutes were most frequently detected in both our results as well as previous examinations, suggesting that bacteria from these phyla are the most prevalent associates of fungal isolates. The average number of detected bacterial associates per fungal isolate from both screens demonstrates that it is common for fungi to harbor multiple bacterial associates, implying that the fungal bacteriome can be quite complex.

We had anticipated that we may discover some potential relationships among bacterial and fungal evolutionary lineages. Initial observations within only 13 *Mortierella* isolates are promising, with at least two different patterns emerging consisting of distinct sets of bacterial taxa (Supplementary Fig. 3). It is thus tempting to speculate that some fungal taxa may harbor one of several 'core' bacteriomes, perhaps dependent on environmental

pressures or its ecological niche. However broader patterns of co-occurring bacteria within related fungi were not apparent. Specific bacterial–fungal associations at the genus level appear more common than generalist associations, suggesting each fungal host harbors a unique bacteriome composed of multiple bacterial associations and that some of these associates are either transient or opportunistic. The absence of any overarching patterns of association may have been impacted by our use of diverse culture collections that utilized different culturing methods, and future explorations into the diverse nature of fungal–bacterial associates will need to tailor methods so as not to impact the natural communities associated with fungi.

While this work provides an important overview and perspective on the diversity of bacterial–fungal associations and the potential complexity of the fungal bacteriome, the underlying mechanisms responsible for these associations remain largely unknown. Elucidating overarching mechanisms responsible for establishing and maintaining bacterial–fungal associations has in the past been hindered in part by the limited diversity and number of described associations, an obstacle addressed directly in this study. Detailed examination of some of these diverse bacterial–fungal partner pairs will help elucidate key genes and pathways that govern bacterial–fungal interactions. Increased knowledge of these underlying mechanisms will be paramount to help predict the biological outcomes of these associations under changing environmental conditions, and their potential impact on ecosystem functioning.

## Methods

**Taxonomic classification of culture collection isolates**. Fungal culture collections from Los Alamos National Laboratory (LANL; New Mexico, USA), University of Neuchâtel (AODJ; Neuchâtel, Switzerland), University of Houston (UH; Houston, USA), and University of São Paulo (BRA; São Paulo, Brazil) were examined in this screen (Supplementary Data 2). These collections were the result of environmental isolations, predominantly from soil-based studies and many of these fungal isolates have been maintained in culture for several years. Mycelia used for sequencing analysis was obtained by inoculating fresh plates of solid media, either malt extract agar (MEA), or potato dextrose agar (PDA), with a small mass of mycelia from stock. Neuchâtel University isolates were inoculated directly onto the media, while isolates from the other three collections were inoculated onto a sterilized sheet of cellophane overlaying the media. These sub-cultured isolates were allowed to grow at room temperature until mycelia covered the entire plate (100 × 15 mm petri dish). All four culture collections were assayed in identical

fashion with the following exceptions: isolates from Houston and São Paulo were grown with the addition of chloramphenicol antibiotic (150 mg/L) to solid media to reduce the growth of exo-bacteria, while isolates from LANL and Neuchâtel were not treated with antibiotics. After sufficient growth was achieved, the cellophane sheet was lifted from the media and the entirety of the fungal mycelia was scraped off and condensed into a pellet for nucleic acid extraction, except for LANL isolates in which case only about a quarter of the fungal growth was used. Fungal biomass of Neuchâtel isolates was harvested by sampling the surface of the colonized agar. The FastDNA™ SPIN kit for Soil (MP Biomedicals, LLC, Solon, OH, USA) was used to extract DNA from the LANL isolates, while the Zymo Quick-DNA Fungal/Bacterial Miniprep Kit (Zymo Research, Orange, CA, USA) was used for Neuchâtel, Houston, and São Paulo isolates. The internal transcribed spacer (ITS) rDNA regions were amplified from the DNA extracts using either ITS5F (5′-GGAAGTAAAAGTCGTAACAAGG-3′; LANL, Houston, São Paulo) or ITS1-F (5′-CTTGGTCATTTAGAGGAAGTAA-3′; Neuchâtel) as forward primer and ITS4 (5′-TCCTCCGCTTATTGATATGC-3′) as reverse primer and the Phusion High-Fidelity DNA polymerase (New England Biolabs, Ipswich, MA, Cat/No. M0531S). Amplified products were then submitted for cleanup and Sanger sequencing (Genewiz, Inc., South Plainfield, NJ, USA). Ambiguous peripheral bases were trimmed using 4Peaks (A. Griekspoor and Tom Groothuis, Nucleobytes, nucleobytes.com) and forward and reverse reads were merged using AliView v1.24[42].

Classification of the ITS amplicons was performed using comparisons to the UNITE 8.3[43] and NCBI BLAST (https://blast.ncbi.nlm.nih.gov/Blast.cg) databases. The UNITE database is a curated database that is accepted as a reliable resource for ITS fungal classification[44], however is less complete than the more encompassing BLAST database. The criteria for identification of fungal taxa required an Expect (E) value of 0.0 and a minimum of 95% sequence identity to an unambiguous top hit (for genus level) in UNITE. When our ITS sequences had matches to multiple closely related fungal UNITE genera with scores that passed our cutoffs (22 out of 294 isolates), the final taxonomic classification was then based on additional BLAST alignments conducted with the NCBI ITS RefSeq database, using the same classification thresholds mentioned above for the original UNITE hits. In addition, a small number of fungal isolates (10) did not meet the identity cutoffs listed, however all but one (AODJ.161.70 which had a best match at 85%) of these isolates aligned to a UNITE database reference with at least 92% sequence identity. Following published guidelines[45] to ensure confident taxonomic classification, the ITS sequences for these 10 isolates were aligned to authenticated and/or published sequences from both the NCBI ITS RefSeq and the complete nucleotide (nt/nr) databases. Top matches were then scrutinized by comparing to other sequences with identical taxonomic classification and authenticated with closely related organisms to increase confidence. Because we cultured all isolates, growth morphology was also considered and no morphologies contradicted our classification analyses.

**Amplification of 16S rDNA from fungal isolates**. Signatures of potentially associating bacteria were examined via amplicon sequencing of the V3-V4 region of 16 S rDNA from the fungal DNA extractions described in the section above. Nested PCR was performed to enhance the bacterial signal. Primers 27F (Lane, 1991)[46] and 907R (Lane, 1991)[46] were used to amplify the V1-V5 region of 16 S rRNA followed by modified versions of primers 341F[47] (5′-CCTACGGGNGGCW GCAG-3′) and 806R[48] (5′-GGACTACHVGGGTATCTAATCC-3′) to further amplify the V3-V4 region prior to sample barcoding. Samples were sequenced on Illumina MiSeq (2 × 300 bp paired end reads). No-template controls and controls for reagent contamination during DNA extraction and library preparation were also included in the sequencing runs for the samples from each culture collection. Blank DNA extractions (controls) were processed in an identical manner to the fungal samples using the reagents from the same lot for both extraction kits (MP Biomedicals and Zymo). These extraction controls were amplified in a manner consistent with the fungal samples and included in the Sanger sequencing submissions. Additionally, all work was conducted in sterilized biological safety cabinets, including separate units in separate labs designated for culturing and molecular work, following strict sterile techniques (e.g. barrier tips, pre- and post-PCR pipettes and workspaces) to limit potential contamination.

**Analysis of 16S rDNA amplicons obtained from fungal isolates**. Raw sequences from all four culture collections were de-multiplexed and were analyzed together as a single pooled group (16S-CC) using QIIME2[49] with DADA2[50] for error modeling and the generation of merged, denoised, chimera-free sequences to produce amplified sequence variants (ASVs). Representative sequences for these ASVs and the ASV table generated by QIIME2 are available in supplemental information (Supplementary Data 6 and Supplementary Data 10). The ASVs were then taxonomically classified using QIIME2's Naïve Bayesian classifier trained on a custom database of the SILVA reference 16S rRNA sequence collection (SSU 138 Ref NR 99) trimmed to the same V3–V4 region produced by the sequencing primers used (Supplementary Data 11). All bacterial genera represented by ASVs identified in the no-template (NTC) or DNA extraction control samples (Supplementary Data 12) were excluded from further analysis within the corresponding culture collection (e.g. if a single *Bacillus* ASV was found in any NTC sample, all *Bacillus* ASVs were excluded from that collection's analysis). Separate controls were used

for each culture collection to try and minimize generalizing contaminants across each collection, as there was not a case where a contaminant ASV or taxa was found in all control samples (likely reflecting the fact that DNA extractions, PCR amplification and sequencing were performed in multiple laboratories). Non-bacterial (mitochondria, chloroplast, and other) ASVs were identified and were excluded from analyses, given our sole interest in bacterial associates. In an effort to reduce redundancy and increase efficiency of downstream analyses given the large number of bacterial ASVs identified (6594), ASVs with identical taxonomic classifications were clustered into 705 bacterial OTUs based on taxonomic identity rather than sequence identity (i.e. all *Bacillus* ASVs were clustered into a single *Bacillus* OTU, regardless of sequence identity). The methods and techniques utilized in this research, such as clustering ASVs into OTUs based on taxonomic identity and the use of DADA2 for error modeling and removal, were selected specifically to eliminate concerns about the artificial inflation of taxonomic richness, by ensuring the removal of spurious sequences and eliminating concerns about erroneous OTU splitting. A total of 22 taxa identified by SILVA were absent from the NCBI taxonomy database when comparing taxonomic classifications and were therefore excluded. OTUs that were not classified at the genus level were also excluded from downstream analyses to simplify taxonomic comparisons.

**Identification of bacterial-specific sequences in fungal genome sequencing projects**. Fungal genome assemblies were downloaded from JGI's Mycocosm data portal using the jgi-query script (https://github.com/glarue/jgi-query). The SRAdb[51] package for R was used to identify the genomic SRA run IDs for each screened fungal isolate using the associated NCBI BioProject number and reads were downloaded using the NCBI SRA Toolkit (Supplementary Data 7). To ensure that all fungal genome sequencing projects used were not under any restrictions for use, only datasets present in the public NCBI SRA database were used. Only runs sequenced using high-throughput technologies (e.g. Illumina and 454) were kept to maximize the probability of capturing bacterial-specific sequences, which were hypothesized to constitute a very small fraction of total sequencing data. All remaining sequencing run data for each fungal genome project were concatenated, meaning in some cases 454 and Illumina data were combined and multiple 454 or Illumina runs for the same fungal isolate were treated as single datasets. This included paired-end Illumina data as well, which was treated as separate single-end data for concatenation. We used the FaQCs[52] module in EDGE[53] to trim bases with quality scores below 30 from the ends of each sequence, and then discarded sequences that met any of the following four criteria: (1) sequence length shorter than 50 bp, (2) sequences with average quality scores <15, (3) sequences with more than one consecutive ambiguous base ('N'), or (4) sequences with a fraction of mono- or di-nucleotide repeats exceeding 65% of the sequence.

In order to enrich bacterial sequences in these fungal projects and reduce computational burden and false positive assignments, fungal sequence data was removed by mapping all reads to the respective fungal genome assembly (assumedly free of bacterial signals). Reads were mapped using the BWA-MEM[54] algorithm and any reads that mapped with 90% or greater similarity to their respective assembly were removed from the quality-filtered sequencing file for each project. The read-based taxonomy classifier GOTTCHA[39] was used to identify and classify bacterial reads from each fungal genome project after the removal of fungal reads. GOTTCHA2 was selected due to its focus on unique genomic signatures that are specific to each bacterial taxon to classify sequencing reads, therefore providing a low false positive rate relative to alternative options. A custom GOTTCHA2 database was created using the NCBI bacterial and viral RefSeq reference genomes (release 89). The default parameters and settings were used for all GOTTCHA2 analyses.

**Phylogenetic comparisons of fungal-derived bacterial sequences**. GOTTCHA2 identified *A. johnsonii*, *B. cereus*, and *E. coli* in multiple fungal hosts from fungal-filtered reads obtained during genome sequencing. Four or more of these fungal projects for each bacterial species were selected for phylogenetic analysis based on the quality of the GOTTCHA2 results (linear coverage and depth of coverage of unique signatures within each project) for the particular bacterial species in question. Once identified, the fungal-filtered reads from these projects were analyzed using PhaME[55] together with several complete reference genomes or reference genome assemblies (per bacterial species) from NCBI RefSeq. Briefly, PhaME automatically selects the most closely related reference genome or assembly to use for alignment of the raw read datasets during the determination of the conserved (core) genome among all input sequence datasets. From this core genome alignment, phylogenetic trees were generated using RAxML and 1000 bootstrap replicates were run for each analysis.

**Design of taxonomic specific FISH probes**. The genus lineage level (SILVA designation D_5) was used to separate the 16S ASVs detected in the *Aspergillus* isolate (LANL.1351.96) into 32 distinct taxonomic groups. OligoMiner[56] was used to design probes for each group following these steps: (1) repetitive sequences, homopolymeric runs and ambiguous bases were masked from all sequences; (2) candidate probes were identified based on provided criteria such as length, melting temperature, and GC content (full parameters provided in Supplementary Data 13); (3) candidate probes were aligned to all off-target sequences including

sequences from the other 31 taxonomic groups and eukaryotic rRNA sequences obtained from SILVA 138.1 LSURef and SSURef; (4) alignments were used to filter out non-specific probes with the recommended LDA model; (5) candidate probes with high abundance kmers and secondary structures were filtered out; (6) the specificity of any remaining probes were compared to 41 *Aspergillus* genomes obtained from NCBI (accessions provided in Supplementary Data 13) using ThermonucleotideBLAST[57] with melting temperature calculated using standard nearest-neighbor thermodynamic parameters at 55 °C; and finally (7) multiplexable checking was performed for qualified probes using ThermonucleotideBLAST.

**Fluorescence in situ hybridization**. Circular growth media disks were prepared by casting 4% gellan gum or phytagel (Cat: P8169-250G, Sigma Aldrich) supplemented with 1X potato dextrose broth (Cat: EW-14200-28. Cole-Parmer) in between two standard microscope slides spaced by two No. 1.5 coverslips. Growth disks were then transferred to a microscope slide fitted with a 65μL Gene Frame (Cat: AB0577, ThermoFisher Scientific). Growth slides were placed in a 100 mm disposable petri dish along with a cap from a 15 mL conical tube filled with water to control humidity. Fungal mycelia from isolates grown for at least 1 week on potato dextrose agar were transferred to the growth media pads and incubated for 3–5 days at 25 °C. It is worthy to note that the different genera of fungi exhibited variable growth phenotypes, and sample preparation for imaging was standardized as best as possible to prioritize thin imaging depths across hyphae.

To prepare samples for FISH, fungal samples were fixed with 4% paraformaldehyde (Cat: 28908, ThermoFisher Scientific) in PBS (Cat: 10010023, ThermoFisher Scientific) buffer at 4 °C overnight directly on the growth disks in the Gene Frames. Samples were washed three times with PBS and cell walls were lysed with a cocktail of 5 mg mL$^{-1}$ lysozyme (Cat: L6876-25G, MilliporeSigma), 5 mg mL$^{-1}$ β-Glucanase from *Trichoderma longibrachiatum* (Cat: G4423-100G, MilliporeSigma), and 500 μg mL$^{-1}$ chitinase from *Streptomyces griseus* (Cat: C6137-25UN, MilliporeSigma) for 1 h at 37 °C. The samples were washed three times with PBS and subsequently dehydrated with a series of ethanol (Cat: T038181000CS, ThermoFisher Scientific) treatments at 50%, 75%, 100%, 75%, 50% and rinsed with PBS for 3 min each at room temperature. Samples stored at −20 °C in 100% ethanol were found to have no change in quality for at least 3 days.

For double-labeled oligonucleotide FISH, Stellaris RNA FISH buffers (Cat: SMF-HB1-10; SMF-WA1-60; SMF-WB1-20, Biosearch Technologies) were used for subsequent hybridization and washing steps. Samples were preconditioned with Stellaris Wash Buffer A supplemented with 30% formamide (Cat: AM9342, ThermoFisher Scientific) for 5 min and replaced by Stellaris Hybridization Buffer supplemented with 30% formamide containing a final 125 nM concentration of each probe. To target endobacteria, a probe pool was generated based on EUB338 targeting the 16S rRNA: EUB338-I (5′-GCT GCC TCC CGT AGG AGT-3′), EUB338-II (5′-GCA GCC ACC CGT AGG TGT-3′), and EUB338-III (5′-GCT GCC ACC CGT AGG TGT-3′) with Cy3 dyes flanking the 5' and 3' ends (Integrated DNA Technologies)[35,58,59]. Universal eukaryote staining was achieved using the 516 region of the 18 S rRNA (5′-ACC AGA CTT GCC CTC C-3′-ATTO 633, EUK516) (Integrated DNA Technologies)[58]. Probes were hybridized for 4 h at 37 °C and then washed three times for 20 min with Stellaris Wash Buffer A at 37 °C. Nuclear staining was performed by incubating 1 μM DAPI (Cat: D1306, ThermoFisher Scientific) in Stellaris Wash Buffer A and for 10 min at room temperature. Excess DAPI was rinsed away with Stellaris Wash Buffer B for 5 min at room temperature. Samples were then rinsed with 2 X SSC buffer (Cat: AM9770, ThermoFisher Scientific) and lastly placed under a No. 1.5 coverslip for imaging.

If direct hybridization of fluorescent probes was insufficient to provide adequate signal, a fluorescence amplification technique was employed. For hybridization chain reaction FISH (HCR-FISH)[36], samples were preconditioned with Probe Hybridization Buffer (Molecular Instruments) for 20 min at room temperature. A pool of target-initiator sequences was proprietarily designed by Molecular Instruments based on the EUB338 target sequence (5′-ACA CUG GAA CUG AGA CAC GGU CCA GAC UCC UAC GGG AGG CAG CAG UGG GGA A′-3′). 1.2 pmol of target probes and 125 nM of EUK516 (5′-ACC AGA CTT GCC CTC C-3′-Quasar 570) (Biosearch Technologies) in Probe Hybridization Buffer was incubated with the samples overnight at 37 °C. Probes were washed 5X with Probe Wash Buffer (Molecular Instruments) for 5 min each and then preconditioned with Amplification Buffer (Molecular Instruments) for 30 min at room temperature. The buffer was replaced with 18 pmol of each HCR amplification probes (Molecular Instruments): B4-Alexa Fluor 647 or B1-Alexa Fluor 488 in Amplification Buffer and incubated overnight at room temperature. Probes were washed 4X with 5X SSCT (5X SSC + 0.1% Tween 20 (Cat: P1379-100ML, MilliporeSigma)) for 5 min each at room temperature. In all, 1 μM DAPI in 5X SSCT was then added to the sample and incubated for 10 min at room temperature and rinsed with 2X SSC three times. A No. 1.5 coverslip was placed on the sample and immediately imaged. Fluorescence microscope images were collected using a Zeiss Axio Observer outfitted with a mercury short-arc lamp, a 1.3 N.A. 100X oil-immersion objective lens, and filters for DAPI, FITC, CY3, and CY5.

Probe pool staining of *Aspergillus* was performed following the protocol outlined above for double-labeled oligonucleotide FISH. Here, Cy3-labeled probes of the probes above were purchased from Biosearch Technologies (probe sequences provided in Supplementary Data 14) or Non-Eub338 (Cy3 - 5′-ACT CCT ACG GGA GGC AGC-3′) was purchased from IDT to stain *Aspergillus* at a final probe concentration of 125 nM in Stellaris hybridization buffer with 30% formamide. DAPI and the Atto-633 labeled EUK516 were used for co-staining at final concentrations of 125 nM.

Confocal microscopy imaging of fungal samples was performed on an Olympus FV3000 laser scanning confocal microscope using a ×100 oil objective lens NA 1.45 and equipped with 405, 488, 561, and 640 nm excitation laser sources and 405/488 and 561/640 dichroic mirrors.

**Image processing**. Image exposures were minimally adjusted to increase brightness across the image using FIJI/ImageJ. For ease of visualization, images were also cropped to magnify bacterial structures within the hyphae. Full lower-magnification images are included in the supplemental information. For confocal image processing FIJI/ImageJ was used to construct 2D maximum projection images for fluorescence channels and additional 3D projections of the images are included as a supplemental file.

**Surveys of previous studies for previously described bacterial–fungal associations**. Literature searches were performed using relevant keywords (bacteria, fungi, bacterial associations, endohyphal bacteria, and endofungal bacteria) to identify research and review articles containing previous descriptions of bacterial–fungal associations. Artificial associations, such as forced associations between bacteria and fungi not isolated from the same environment or associations involving genetically engineered strains, were excluded, as well as co-occurrence studies involving multiple fungi. Our findings are summarized in Supplementary Data 3 and include links to the references where each association is described. Due to a number of challenges such as inconsistent usage of keywords in these publications, descriptions of associations at varying taxonomic levels and descriptions of associations appearing exclusively in figures, tables and/or supplementary information, a complete summary of previous descriptions would require a separate, and independent effort. Nevertheless, the data compiled in Supplementary Data 3 represents to our knowledge the first attempt to compile a comprehensive series of previously published descriptions of associations between bacteria and fungi.

**Reporting Summary**. Further information on research design is available in the Nature Research Reporting Summary linked to this article.

## Data availability
The ITS sequences used for classification of the fungal isolates have been deposited at NCBI GenBank and the unprocessed 16S amplicon sequencing data obtained from these isolates have been deposited in the NCBI SRA database (BioProject accession number: PRJNA738181). All other source data is contained within the supplemental material, which are available through figshare[60]: https://doi.org/10.6084/m9.figshare.c.5582283.v4.

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

## Acknowledgements

This study was supported by the U.S. Department of Energy, Office of Science, Biological and Environmental Research Division, under award number LANLF59T. This work was performed, in part, at the Center for Integrated Nanotechnologies, an Office of Science User Facility operated for the U.S. Department of Energy (DOE) Office of Science.

## Author contributions

A.J.R., G.L.H., D.P.M., K.W.D., S.B., D.F.R., J.D.Y., P.J. and P.S.G.C. contributed to the planning and designing of the overall study and analyses. A.J.R., G.L.H., D.P.M., J.M.K., L.G., F.P., A.L., D.B., A.E., M.B., C.P., V.H., S.L., T.J., A.O.Z. and H.N.N. contributed to the experimental work including culturing of fungal isolates, DNA extractions, and PCR amplification. D.P.M. and J.H.W. performed the microscopic experiments and analyses. A.J.R., P.J. and P.S.G.C. contributed to the bioinformatic and phylogenetic analyses. L.J.G., S.B., D.F.R. and P.J. contributed fungal isolates used in these analyses. A.J.R., E.S.L. and G.C. contributed to figure and graphic design. A.J.R., G.L.H., D.P.M., P.J. and P.S.G.C. wrote and reviewed the manuscript. All authors read and approved the final manuscript.

## Competing interests

The authors declare no competing interests.
