## [Transparent Peer Review File · Communications Biology]

Reviewers' comments:

Reviewer #1 (Remarks to the Author):

Thank you for the opportunity to review this paper, which focuses on detection and description of bacteria associated with diverse fungi. The authors first use 16S rRNA sequencing on the Illumina platform to detect and identify bacteria in representative fungi from culture collections. They also mine publicly available fungal genome sequences for evidence of bacteria. Finally they use visualization methods (FISH, microscopy) to observe bacteria in association with focal fungi. The authors report a very high richness of bacteria associated with fungi, particularly as detected by their 16S screening. The work expands the current perspective on bacterial associates of fungi beyond the Dikarya, Glomeromycota/tina and Mucoromycota/tina, especially through the genome-mining approach. Overall the authors conclude that the bacteriomes of fungi contain unprecedented bacterial diversity.

I am excited by the authors' complementary approaches and the scope of this work. The microscopy is a terrific addition. I agree wholeheartedly with the take-away that bacteria associated with fungi merit much more attention and I appreciate that the authors explored that diversity from various angles. The discovery of ever-greater diversity of bacteria associated with fungi will set the stage for numerous new questions about fungal and bacterial phenotypes, virulence, ecology, interactions, and evolution.

While I am enthusiastic about the approaches and focus of the paper, I also identified several concerns. I describe them below. In brief, they are concerns about:

- (1) the conclusions given aspects of the methods and data;
- (2) limited inference given the rich metadata associated with focal fungi;
- (3) limited information in figures/data presentation;
- (4) other matters, including spelling errors in fungal names, inclusion of Archaea when referring to bacteria, etc.

Most generally, I am concerned about the implications of 'associations' as used in the paper. Truly what the authors did was discover bacterial DNA signatures in fungal genomic DNA from cultures; bacterial DNA signatures in fungal genome data; and in some bacteria/fungal pairs, they imaged to visualize associations. Only in the last do we have evidence of 'association' rather than 'co-isolation' or 'contamination.' The authors speak broadly to their terminology late in the paper, but it feels rather implied throughout, up to that point, that they are thinking about biologically interacting/associated/co-isolated organisms. I would suggest the authors step back and simply refer to detection of DNA signatures + images of associations in some cases.

With that in mind I strongly encourage the authors not to orient their paper around addressing neglected epiphythal bacteria (authors' statement in the abstract) as they have no insights from the majority of the work presented here re. epiphythal or endohyphal status (or contaminant status in cultures, DNA preps for genome sequencing, etc. etc.).

In addition here are my main concerns:

1. Concerns about conclusions given aspects of the methods and data

I am concerned about the conclusion of 'unprecedented diversity' of bacteria without several additional aspects of quality control, details of methods, and transparency.

1a. Concerns about contamination. I was very happy that the authors mentioned negative controls (extraction and reagent blanks) in their 16S survey. To their credit, they removed ASVs that were found in those control samples prior to analyzing their 16S data. The authors' case would be strengthened here by greater transparency and detail. To that end please:

(i) provide more information about these controls (were they from all kits, both kit manufacturers, and prepared in a manner consistent with the nested PCRs?);

(ii) provide a supplement with read numbers and ASVs from these thorough negative controls, so that we know if they were (for example) prevalent and the 'true data' were, for example, a minority of the reads in a given prep;

(iii) consider whether the negative controls could be used more effectively by collapsing into OTU first (all negative controls and real data), then removing OTUs in both, rather than removing ASVs observed in the negative controls prior to collapsing the remaining ASVs from the real data into OTU.

(iv) state in the methods the % sequence similarity used for OTU delimitation (presumably 97%?). The algorithm should also be stated, as these differ in outputs for the same input.

Moreover, (v) the authors' case would be strengthened further if the authors would please explain additional steps taken to limit contamination, e.g., working with barrier tips, distinct pipettes for pre- and post-PCR samples, working in sterile spaces, etc. If the conclusion is 'unprecedented biodiversity' we have to know that the work ruled out every possible false source of said biodiversity.

1b. Potential for false signal due to nested PCR approach, sequencing error, etc. The authors used a nested PCR approach to amplify the bacterial signal in their DNA extractions prior to Illumina sequencing. Nested PCRs can be problematic in generating chimeras and it is understood in general that nested PCRs for biodiversity discovery 'must be validated by other methods' - especially for rare taxa (just one example, <https://journals.plos.org/plosone/article?id=10.1371/journal.pone.0132253>). There is no evidence readily visible that chimeras were removed from the data set. They would artificially increase richness and yield previously unknown taxa if present. Therefore, (i) please explain further that chimera-checking was done and examine whether artifactual diversity could be generated due to the nested PCR approach.

With regard to sequencing errors, I would direct the authors also to <https://pubmed.ncbi.nlm.nih.gov/19725865/>. While the sequencing technology used here is different, the principles laid out there (of homopolymers, unresolved bases, and the need for stringent quality thresholds/etc.) remain important. Here, (ii) much more detail about quality control is needed — at a bare minimum, sequence length cutoffs, max ee values, etc.

(iii) The authors also could strengthen their case considerably by providing evidence from a mock community that sequencing error rates are not artificially increasing richness of bacteria in their samples. If they did not sequence a mock community then perhaps careful scrutiny of mitochondrial sequence data from several individual samples would allow them to understand error rate in their data.

(iv) I was surprised that chloroplast sequences needed to be excluded from fungal genomic DNA (methods)?

(v) By necessity (because they are using published genome sequences) the authors could not implement any negative controls for the bacterial sequence survey from the fungal genome data. The authors note that perhaps the lesser prevalence of bacterial sequences in the genome survey reflects the low quantity of bacterial 16S in those preparations relative to the selectively amplified 16S data

from their Illumina work. While unlikely another interpretation (which can be ruled out by addressing the points raised here) is that the Illumina work was compromised by exogenous bacteria. The authors should please offer some thoughts on this matter.

(vi) For comparisons of the bacterial data from fungal genome sequencing vs. the Illumina survey, are differences in error rate or other matters relevant and addressed sufficiently to support the comparisons made in Fig. 1?

(vii) The authors note that the genome screening approach allows detection of more than 16S for bacterial DNA sequences captured with fungal genome data. Please clarify how cases of horizontal gene transfer (HGT; e.g., DNA of bacterial origin now appearing as part of the fungal genome due to HGT) can be distinguished from evidence for 'bacterial associates.'

(viii) I believe the final analyses of the bacteria from the Illumina survey are based on OTUs, generated from ASVs. Sometimes the same OTU appears in multiple fungi. This seems surprising if the fungi were, for example, from different regions or habitats, processed by different collection dates or labs, etc. Is it possible that these differ at the ASV level? It seems unlikely that the same bacterial ASV would appear in samples from Europe vs. North America (for example) unless the 16S sequence data can't resolve them finely enough to differentiate close relatives (i.e., things are lumped somehow) or there were contamination in the processing. This can be relieved as a concern by addressing the points above.

1c. Naming of taxa. The authors speak to finding many new genus-level pairings of fungi and bacteria. In doing so they rely on estimates of genus-level identification. This is problematic without further qualification for several reasons, including matching to unnamed fungi (very common with field-collected samples such as soil fungi), obtaining multiple top matches to different taxa that are equal quality, the non-phylogenetic matching criteria, etc. Moreover, no identity thresholds or other identification criteria are listed. (i) These should all be addressed.

(ii) I was also surprised by the large number of Ns in the ITS sequences for fungi — seems very high in our experience and it makes the sequence data suspect. Please explain and validate the sequence quality overall to provide greater confidence.

(iii) Sometimes previous findings are not quite represented correctly. For example the authors state that *Penicillium* was not previously known to associate with bacteria, but wasn't such an association shown in the case of (for example) an endophyte in the *Penicillium* group by Hoffman and Arnold 2010? Also, please see: GhodsSalavi B, Svenningsen NB, Hao X et al. A novel baiting microcosm approach used to identify the bacterial community associated with *Penicillium bilaii* hyphae in soil. *PLoS One* 2017;12. DOI: 10.1371/journal.pone.0187116.

1d. Transparency. I must have missed it, but could not find evidence that DNA sequences generated in this study were deposited in a publicly accessible database (the ITS sequences are in a supplement; what about the bacterial data). (i) Could accession numbers for the short-read archive or other deposition be included?

(ii) Please state that all JGI genome used were fully public and not under any restrictions for use, per <https://jgi.doe.gov/user-programs/pmo-overview/policies/#data-util> (note, e.g., pre 2018 proviso).

(iii) It would be helpful to understand more about the metadata of the focal fungi, per (2), below.

2. Limited inference given the rich metadata associated with focal fungi

As written the paper entirely neglects hypotheses or predictions as a conceptual framework. While I

am a fan of 'discovery' papers it seems that the authors could strengthen this particular work with attention to hypotheses/predictions that guide the analyses. Specifically, (i) I believe the authors must have gone into the work with some predictions based on their scholarship and their selection of focal strains. It would relieve the concern about contamination, for example, if the authors found that the geographic origin of fungi, processed in the same lab, nonetheless was key in structuring associated bacterial communities. The work could move from more of a 'fishing and phenomenological paper' to something more insightful, given the wealth of data the authors have generated. This relates to 3, below. I'm definitely not suggesting 'straw man' hypotheses, but structure to the thinking presented here could greatly strengthen the inference.

3. Concerns about figures and related data presentation. Fig. 1, it's up to the authors of course, but I wonder if this is a key finding of the paper — given the methodological points above and the general limitations of Venn diagrams. I struggled to see the details in Fig. 2 (so small).

4. Other matters

Why are these archaea listed as bacteria in Fig. 3? Crenarchaeota, Euryarchaeota

Please check the spelling of fungal names — Phyllosticta is not correct (Phyllosticta). I am surprised the fungal culture collections included Lobaria, which is a lichen-forming fungus.

Sometimes the Results section included Discussion-type material ("These overlaps in the niche and trophic mode of the fungal hosts could be one possible explanation for the phylogenetic similarity found between their bacterial associates. These results collectively indicate that diverse fungi can harbor phylogenetically-related bacteria, which suggests a functional basis for these interactions.")

Reviewer #2 (Remarks to the Author):

The research submitted by Robinson and collaborators focus on the exploration of bacterial associates of fungi. To gain a broader overview of these fungal-bacterial associations, authors performed two different strategies:

1. They used amplicon sequencing of the 16S rDNA gene from total DNA of 294 fungal strains preserved in four fungal collections (Houston, Los Alamos, São Paulo and Neuchâtel).
2. They searched for bacterial signatures in 408 fungal genome projects available at the MycoCosm database.

Some of the newly discovered and known fungal-bacterial interactions (from 1) were examined in more detailed using microscopy (FISH or HCR-FISH).

Authors main findings are that potential bacterial associates of fungi are more widespread and diverse than anticipated, but no clear patterns were observed in terms of the number of bacterial associates and/or the bacterial lineages associated with fungal hosts (similar to findings by Hoffmann & Arnold AEM 2010). Altogether, the results presented here support the idea that fungal biology, and potentially the biology of plant and animals that harbor also a mycobiota, may be affected by these associations.

Despite the study is of interest and presents a broad investigation, the major results are, in my opinion, not concisely and well presented (in an informative way). Thus, the conclusions lack support and clear evidence.

Authors, although provided with ITS fungal sequences or (nearly) full genome data, do not present in a clear way the fungal taxa included in the study (a phylogeny) and the sample size of each group (Phylum, family, genus) for each of the two approaches. Authors could follow to some extent the way Myers et al., mBio 2020 presented the fungal taxa screened and the “potential bacterial (instead of viral)” associates found.

For the reader it is difficult now to really comprehend and see (in figures!) which fungal groups were examined and which bacterial associates were found (the likely best representation of this was Supplementary Figure 6, in my opinion).

Also, sequence data (fungal ITS sequences and 16S barcoding) seems not to be deposited in public databases, as no accession numbers and/or bioprojects are referenced in the main text. Please include.

I have some concerns about the methodology used for the 16S-CC screen. It is unfortunate that protocols were, to some extent, differently performed by the four participating Labs. Some used chloramphenicol for the subculturing of the fungi, some did not. Chloramphenicol is a broad spectrum antibiotic, so it definitely can affect the “bacteriome” associated with a fungus. This has been already reported in *Mucor hiemalis* (Schulz-Bohm et al., Fungal Genet Biol. 2017). Did this treatment affect the number and taxa of the potential bacterial associates found? Authors should quantitatively account for this aspect. Have chloramphenicol-treated fungi on average less bacterial OTUs than non-treated? Have they different bacterial taxa when compared to other fungi of the same genera that were left untreated?

By reading the methods, I was unable to clearly identify if authors have biological replicas of the 16S barcoding of some (or most) of their fungal samples. For the microbiome of plants and animals, 3 biological replicas are common. Do authors have biological replicas? were biological replicas similar among them? were negative controls also included (known bacteria-free fungi such as some reported strains of *R. microsporus*)? The barcoding data allows not only to identify which taxa is present, but also its relative abundance. I wonder why authors do not present this information. The analysis of this data could be further used to infer which symbionts are more abundant in a fungal host, and which ones could be secondary symbionts. This piece of information could be later validated using FISH with specific probes. Moreover, authors could also use some tools of microbial ecology (NMDS, Permanova/AnoSim, etc.) to further support their findings, especially to test if the fungal host and its lineage (or the use of chloramphenicol and/or the ecological niche) is a factor influencing bacterial community assembly.

FISH and HCR-FISH were only used in the context of the study to identify bacterial cells in fungi, but not to show any example of a complex/diverse bacterial community within a host. Authors report that some of the screened fungi have up to 100 OTUs and an average of 34 per strain (lines 122-3). Unfortunately, these numbers can not be directly evidenced from their microscopic studies. Based on the micrographs displayed in Figure 3, I was unable to observe bacterial cells in the outer surface of fungal cells (epibionts). Authors claim that their study incorporates both external and internal associates of fungi, but I am afraid that the methods used do not allow them to separate these two groups.

It would have been also desirable that fungal cultures used for DNA extraction were derived from single spores (monosporic cultures) to make sure associations are tight, and not only circumstantial. The finding that all strains screened in the 16S-CC were associated with at least one bacterial associate sounds suspicious to me. In previous published screenings the frequency of finding bacterial symbionts in several fungal hosts has never been 100% (see please Hoffmann & Arnold AEM 2010, Desiró et al. ISME J 2014, Toomer et al. Mol Ecol 2015, Dolatabadi et al. Fungal Biol 2016, Okrasińska et al. AEM 2021, etc.). Thus, it is very likely that authors are reporting spurious associations.

With respect to the genome-based strategy (or BSS), one of the key aspects missing is, in my opinion, to show a quantitative overview of the bacterial signatures found. From reading the actual manuscript,

it is hard to visualize:

1. From each fungal genome analyzed, how many sequences (contigs and their lengths) were putatively associated with bacteria? Which proportion do they represent from the total DNA? What is the relative abundance of each bacterial group associated with a fungal host?...

In this approach, I liked very much that authors investigated SNPs in the bacterial signatures associated with *B. cereus*, *E. coli* and *A. johnsonii* from several fungal hosts. These results do show that similar bacterial symbionts are putatively associated with diverse fungal taxa, and that these bacterial strains are probably similar to bacteria that occupy other ecological niches. In Figure 5, it would be desirable that authors add an icon to specify the fungal phylum of occurrence and/or the ecological niche of the bacterium.

I strongly suggest authors to think in ways in which both approaches (16S-CC and BSS) could render similar, informative and concise figures that allow readers to easily make comparisons and reach conclusions.

Most actual figures are either too simple (figure 1) or too complex (figure 2, 4) to deliver valuable information.

I also recommend authors to discuss in more detail the limitations of their study, and to provide the most promising perspectives gained through the analyses done. Are there specific fungal genera that would enable a broad understanding of BFI? Or studies should focus more on the same ecological niche? Are 35 bacterial OTUs stably associated with "axenic" fungi?

Reviewer #3 (Remarks to the Author):

The manuscript by Robinson and colleagues reports results of two surveys targeting bacteria associated with fungi: (1) empirical investigation of 294 fungal isolates from four culture collections (16S-CC screen) and (2) computational examination of 408 fungal genomes sequenced by JGI (BSS). While both efforts yielded evidence of numerous bacterial taxa associated with focal fungi, which is a source of valuable information, presentation and interpretation of these data requires adjustments, as discussed below:

1. The study is a "fishing expedition" devoid of any ecological context. Therefore, its findings are at best observations that could be used to formulate hypotheses for further studies. In particular, finding DNA sequences of *Bacillus cereus*, *E. coli* and *Acinetobacter johnsonii* in multiple fungal genomes should not be over-interpreted. The prudent hypotheses here would be that these bacteria are indeed associated with their respective hosts. If these hypotheses are supported by empirical tests of living fungal cultures, then patterns of associations could be examined.

2. I have a problem with framing of the rationale for this study (lines 53-57), suggesting that community-level ecological studies of fungal-bacterial interactions are not informative. Most of the citations used to support this notion are review papers because community-level field studies are in their infancy and these ecological data are still limited. However, once such studies start yielding data, they will be vastly superior to the approach taken by the authors.

3. Line 97. Please list these collections here and describe briefly how sequence data were generated. With M&Ms in the back, it is difficult to follow the Results.

4. Line 206 and 400. The authors keep referring to fungal hyphae/mycelium as tissues. Please correct.

5. Lines 283-288. Were Mycetohabitans, Mycoavidus and Mycoplasma-related bacteria not detected?

6. I was intrigued by the networks of fungal-bacterial associations presented in Figure 4. However,

upon closer inspection, I realized that only panels a and c are informative; panel b is vastly confusing and needs to be explained better. Perhaps redrawing this network as two separate networks: 16S-CC and BSS would work better. It would be also useful to know what tool was used to draw this network.

7. In the References, please make sure that journal titles are capitalized.

Reviewers' comments:

Reviewer #1 (Remarks to the Author):

Thank you for the opportunity to review this paper, which focuses on detection and description of bacteria associated with diverse fungi. The authors first use 16S rRNA sequencing on the Illumina platform to detect and identify bacteria in representative fungi from culture collections. They also mine publicly available fungal genome sequences for evidence of bacteria. Finally they use visualization methods (FISH, microscopy) to observe bacteria in association with focal fungi. The authors report a very high richness of bacteria associated with fungi, particularly as detected by their 16S screening. The work expands the current perspective on bacterial associates of fungi beyond the Dikarya, Glomeromycota/tina and Mucoromycota/tina, especially through the genome-mining approach. Overall the authors conclude that the bacteriomes of fungi contain unprecedented bacterial diversity.

I am excited by the authors' complementary approaches and the scope of this work. The microscopy is a terrific addition. I agree wholeheartedly with the take-away that bacteria associated with fungi merit much more attention and I appreciate that the authors explored that diversity from various angles. The discovery of ever-greater diversity of bacteria associated with fungi will set the stage for numerous new questions about fungal and bacterial phenotypes, virulence, ecology, interactions, and evolution.

While I am enthusiastic about the approaches and focus of the paper, I also identified several concerns. I describe them below. In brief, they are concerns about:

- (1) the conclusions given aspects of the methods and data;
- (2) limited inference given the rich metadata associated with focal fungi;
- (3) limited information in figures/data presentation;
- (4) other matters, including spelling errors in fungal names, inclusion of Archaea when referring to bacteria, etc.

Most generally, I am concerned about the implications of 'associations' as used in the paper. Truly what the authors did was discover bacterial DNA signatures in fungal genomic DNA from cultures; bacterial DNA signatures in fungal genome data; and in some bacteria/fungal pairs, they imaged to visualize associations. Only in the last do we have evidence of 'association' rather than 'co-isolation' or 'contamination.' The authors

speak broadly to their terminology late in the paper, but it feels rather implied throughout, up to that point, that they are thinking about biologically interacting/associated/co-isolated organisms. I would suggest the authors step back and simply refer to detection of DNA signatures + images of associations in some cases.

We acknowledge making a distinction between ‘associates’, ‘co-isolates’ and ‘contamination’ can be quite difficult, even in the case of direct visualization, which is why we describe the associations as ‘putative’ or ‘potential’ and generally use phrasing indicating our lack of absolute certainty where appropriate. These modifiers now appear more frequently throughout the text to address this concern. Additionally, we use as precedent the efforts of studying animal microbiomes, where many taxa are generally reported, but in most cases not further investigated. An example of this would be skin microbiomes which contain transient subpopulations.

Additionally, we added the following statement in the introduction to address the concern by the reviewer (lines 56-59): “Focusing on fungal isolates has the aforementioned benefits, but distinguishing between true biological associations and other factors resulting in the detection of a bacterial signature (e.g. co-isolation or contamination) can still be difficult without detailed and time-intensive investigations of each putative association.”

With that in mind I strongly encourage the authors not to orient their paper around addressing neglected epiphythal bacteria (authors’ statement in the abstract) as they have no insights from the majority of the work presented here re. epiphythal or endohythal status (or contaminant status in cultures, DNA preps for genome sequencing, etc. etc.).

This was not what we had intended for readers to interpret, and have thus removed this entire statement from the abstract.

1. Concerns about conclusions given aspects of the methods and data

I am concerned about the conclusion of ‘unprecedented diversity’ of bacteria without several additional aspects of quality control, details of methods, and transparency.

We have included additional information in the manuscript to address the reviewer’s concerns, as outlined in the specific points below.

1a. Concerns about contamination. I was very happy that the authors mentioned negative controls (extraction and reagent blanks) in their 16S survey. To their credit,

they removed ASVs that were found in those control samples prior to analyzing their 16S data. The authors' case would be strengthened here by greater transparency and detail. To that end please:

(i) provide more information about these controls (were they from all kits, both kit manufacturers, and prepared in a manner consistent with the nested PCRs?);

The following statement has been added to the Methods section to address this concern (lines 514-517): “Blank DNA extractions (controls) were processed in an identical manner to the fungal samples using the reagents from the same lot for both extraction kits (MP Biomedicals and Zymo). These extraction controls were amplified in a manner consistent with the fungal samples and included in the Sanger sequencing submissions.”

(ii) provide a supplement with read numbers and ASVs from these thorough negative controls, so that we know if they were (for example) prevalent and the ‘true data’ were, for example, a minority of the reads in a given prep;

In general, prevalence of apparent contaminants is near impossible to ascertain without spike-in controls at known concentrations in each sample, given that the interpretation of amplicon sequencing data is based on relative abundances. It is also worth reiterating that separate controls were utilized for each collection and comparisons of ASV relative abundance between controls and samples should be conducted separately within each collection.

In order to allow the readers to independently assess the impact of the negative control samples in the dataset, we have now included the ASV table from the QIIME2 analysis (Supplementary Table 4), the representative sequences for each ASV in FASTA format (Supplementary Data 1), the taxonomic assignments and their confidence for each representative sequence (Supplementary Table 8), and the ASVs found in the control samples as well as their counts (Supplementary Table 9).

(iii) consider whether the negative controls could be used more effectively by collapsing into OTU first (all negative controls and real data), then removing OTUs in both, rather than removing ASVs observed in the negative controls prior to collapsing the remaining ASVs from the real data into OTU.

We thank the reviewer for this comment and had actually utilized a similar, but slightly more aggressive approach to handling bacterial signals found in our controls. We have also modified language in the text to better explain our procedures.

Our strategy for rolling up ASVs into OTUs was based on taxonomic identity (rolling up to the genus level from species) rather than sequence identity (used only for clustering into ASVs), and we added a statement to make this more understandable in the Methods (lines 542-547).

The following statements were added to the methods section to better explain the approaches utilized to exclude potential contaminants (lines 531-538): “All bacterial genera represented by ASVs identified in the no-template (NTC) or DNA extraction control samples (Supplementary Table 9) were excluded from further analysis within the corresponding culture collection (e.g. if a single *Bacillus* ASV was found in any NTC sample, all *Bacillus* ASVs were excluded from that collection’s analysis). Separate controls were used for each culture collection to try and minimize generalizing contaminants across each collection, as there was not a case where a contaminant ASV or taxa was found in all control samples (likely reflecting the fact that DNA extractions, PCR amplification and sequencing were performed in multiple laboratories).”

(iv) state in the methods the % sequence similarity used for OTU delimitation (presumably 97%?). The algorithm should also be stated, as these differ in outputs for the same input.

As mentioned above, because our strategy for rolling up ASVs into OTUs was based on taxonomic lineage assignment rather than sequence identity, this OTU delimitation criterion does not apply. However, we have added a statement to make our protocol more clear in the Methods, as mentioned above.

Moreover, (v) the authors’ case would be strengthened further if the authors would please explain additional steps taken to limit contamination, e.g., working with barrier tips, distinct pipettes for pre- and post-PCR samples, working in sterile spaces, etc. If the conclusion is ‘unprecedented biodiversity’ we have to know that the work ruled out every possible false source of said biodiversity.

We added the following statement to the Methods section (lines 517-520): “Additionally, all work was conducted in sterilized biological safety cabinets, including separate units in separate labs designated for culturing and molecular work, following strict sterile techniques (e.g. barrier tips, pre- and post-PCR pipettes and workspaces) to limit potential contamination.”

1b. Potential for false signal due to nested PCR approach, sequencing error, etc. The authors used a nested PCR approach to amplify the bacterial signal in their DNA extractions prior to Illumina sequencing. Nested PCRs can be problematic in generating chimeras and it is understood in general that nested PCRs for biodiversity discovery 'must be validated by other methods' - especially for rare taxa (just one example, <https://journals.plos.org/plosone/article?id=10.1371/journal.pone.0132253>). There is no evidence readily visible that chimeras were removed from the data set. They would artificially increase richness and yield previously unknown taxa if present. Therefore, (i) please explain further that chimera-checking was done and examine whether artifactual diversity could be generated due to the nested PCR approach.

We agree with the reviewer that having a nested PCR approach might lead to potential issues for the interpretation of the results. This is the main reason why we have restricted our analysis to presence/absence, rather than quantitative assessment of diversity based on relative abundances. However, this approach was necessary given the extremely low relative amount of DNA from the associated bacteria, relative to the DNA from the fungal host. Regarding the elimination of chimeras, we have altered our methods description (lines 524-526) regarding the use of DADA2 package in the QIIME2 workflow to make it clear that it was not only used for error modeling, but also denoising and the removal of chimeras prior to ASV generation. DADA2 (<https://www.nature.com/articles/nmeth.3869>) is one of the most popular bioinformatic standards used for processing amplicon data because it addresses many of the concerns that this reviewer has raised. A recent review (2020) entitled "Comparing bioinformatic pipelines for microbial 16S rRNA amplicon sequencing" ([10.1371/journal.pone.0227434](https://doi.org/10.1371/journal.pone.0227434)) states: "We found DADA2 to be the best choice for studies requiring the highest possible biological resolution (e.g. studies focused on differentiating closely related strains)." We believe the use of this software adequately addresses all of the concerns raised by the reviewer.

In order to make it clear that these concerns and issues were considered when conducting our research, we also included a new statement in the Methods (lines 544-547): "The methods and techniques utilized in this research, such as rolling up ASVs into OTUs based on taxonomic identity and the use of DADA2 for error modeling and removal, were selected specifically to eliminate concerns about the artificial inflation of taxonomic richness, by ensuring the removal of spurious sequences and eliminating concerns about erroneous OTU splitting."

With regard to sequencing errors, I would direct the authors also to <https://pubmed.ncbi.nlm.nih.gov/19725865/>. While the sequencing technology used here is different, the principles laid out there (of homopolymers, unresolved bases, and

the need for stringent quality thresholds/etc.) remain important. Here, (ii) much more detail about quality control is needed — at a bare minimum, sequence length cutoffs, max ee values, etc.

In addition to using the widely accepted DADA2 package for the 16S screen (see above), which already performs error analysis and correction, we have also added some of our quality control information for the bioinformatic screen of fungal genome projects (lines 566-570): “We used the FaQCs module in EDGE to trim bases with quality scores below 30 from the ends of each sequence, and then discarded sequences that met any of the following four criteria: 1) sequence length shorter than 50 bp, 2) sequences with average quality scores less than 15, 3) sequences with more than one consecutive ambiguous base ('N'), or 4) sequences with a fraction of mono- or di-nucleotide repeats exceeding 65% of the sequence.”

(iii) The authors also could strengthen their case considerably by providing evidence from a mock community that sequencing error rates are not artificially increasing richness of bacteria in their samples. If they did not sequence a mock community then perhaps careful scrutiny of mitochondrial sequence data from several individual samples would allow them to understand error rate in their data.

While we agree that the utilization of mock communities could help with more accurate assessment of sequencing error rates, DADA2 is commonly used for its ability to reduce the inclusion of spurious sequences and increase the confidence in any identified variants for amplicon data analysis. A published evaluation of DADA2 using diverse mock communities and Illumina amplicon datasets revealed that “These comparisons show that DADA2 is more accurate than other methods. DADA2 resolves fine-scale variation better than the method currently considered most robust for that task, but it outputs fewer incorrect sequences than the best OTU method. The precision of DADA2 improves downstream measures of diversity and dissimilarity and could potentially allow amplicon methods to probe strain-level variation.”

<https://www.nature.com/articles/nmeth.3869>).

The utilization of this tool reduces the inclusion of any spurious sequences, thereby already minimizing concerns of inflated richness. Furthermore, it is important to consider that these *in silico* error correction techniques were applied when reporting ASVs, where error rates are an important consideration as a single incorrect substitution could result in over-representation of the diversity of certain taxonomic groups. We present the data based on rolling up the 6,594 ASV assignments to the genus level (546) or higher taxonomic level (159), as opposed to presenting the data at the ASV level, which

further reduces the chances of overinflating richness by eliminating erroneous OTU splitting due to minor sequencing errors.

In our bioinformatic screen of fungal genome projects, we utilized GOTTCHA2, which uses genomic signatures unique to each genome at any given taxonomic level to confidently classify bacterial reads that uniquely match that taxon, while accounting for typical sequencing errors within its analysis. This tool, as well as the others used in this study have all been tested and evaluated on mock or simulated as well as real datasets, and are considered highly accurate in their estimations of richness.

(iv) I was surprised that chloroplast sequences needed to be excluded from fungal genomic DNA (methods)?

We were also somewhat surprised, but further investigation indicated the detected sequences are identical to plant chloroplast (or plastid) sequences, represented by multiple reads, and are from plants that are neither grown or sequenced in our facilities. Other papers that have employed similar methods (16S from fungal hyphae) have also found chloroplast sequences (<https://www.nature.com/articles/s41396-020-0674-7>) (within the 'Bioinformatic and statistical analysis' section).

(v) By necessity (because they are using published genome sequences) the authors could not implement any negative controls for the bacterial sequence survey from the fungal genome data. The authors note that perhaps the lesser prevalence of bacterial sequences in the genome survey reflects the low quantity of bacterial 16S in those preparations relative to the selectively amplified 16S data from their Illumina work. While unlikely another interpretation (which can be ruled out by addressing the points raised here) is that the Illumina work was compromised by exogenous bacteria. The authors should please offer some thoughts on this matter.

In case there was any misinterpretation by the reviewer, we do not argue about a lower quantity of bacterial 16S or genomic DNA in the the whole genome sequencing project preparations, but rather, as further discussed above and below, a large relative abundance difference between bacterial cells and genomic DNA vs fungal cells and genomic DNA for those sequencing projects. The JGI, arguably the world's premier fungal isolate sequencing center, has strict guidelines and quality controls in place before accepting nucleic acid materials and works with the submitters to ensure high quality material is received (which was reported as corresponding to axenic cultures by the submitters) and that their genomes are in fact screened for bacterial and other contaminants prior to data submission to NCBI. The genome projects screened were all sequenced by JGI, which gives us confidence in their standardized procedures. No

patterns were observed to suggest contamination across all JGI genomes, and similarly genomes originating from the same project or laboratory also did not show common bacterial signatures, meaning that any further potential contamination would have to be sample prep specific. For example, we have examined fungal samples submitted from the same lab (<https://doi.org/10.1371/journal.ppat.1003037>), and while all were found to harbor bacterial signatures, taxonomic analyses indicate distinct bacterial profiles for each isolate, adding further support for their designation as associates rather than contaminants originating from the protocols, lab or the sequencing center.

(vi) For comparisons of the bacterial data from fungal genome sequencing vs. the Illumina survey, are differences in error rate or other matters relevant and addressed sufficiently to support the comparisons made in Fig. 1?

As described above, the techniques used in both screens accounted for sequencing error rate and other typical concerns when analyzing both amplicon and metagenomic data. The manuscript is now transparent about both the methods of taxonomic

assignment, which are appropriate for both groups of organisms, and the confidence of those taxonomic assignments. The comparisons in Figure 1 are all conducted at the genus level, rather than the species or ASV level, further helping to alleviate any concerns related to artificial inflation of richness that may result when examining NGS data.

(vii) The authors note that the genome screening approach allows detection of more than 16S for bacterial DNA sequences captured with fungal genome data. Please clarify how cases of horizontal gene transfer (HGT; e.g., DNA of bacterial origin now appearing as part of the fungal genome due to HGT) can be distinguished from evidence for 'bacterial associates.'

The method applied was designed to avoid this type of mis-interpretation. In the Methods section (lines 572-574) we mention: "In order to enrich bacterial sequences in these fungal projects and reduce computational burden and false positive assignments, fungal sequence data was removed by mapping all reads to the respective fungal genome assembly (assumedly free of bacterial signals)." This means that if the fungal host genome experienced HGT from a bacterial associate, and it was correctly represented in the representative assembly, reads mapping to these regions would be excluded prior to classification as potential associates. Moreover, if spurious reads of HGT origin were unable to be mapped into the assembled fungal genome, these reads could be identified as bacterial (given sufficient sequence identity for alignment) but would have a characteristic pattern, originating from the same local part of the bacterial genome (pileup), which is not what we found.

The software used to identify bacterial genome signatures in the fungal genome screening (GOTTCHA2) utilizes signatures throughout all available bacterial genomes and pan-genomes to confidently identify and classify reads that uniquely map only to a particular taxon. Genomic loci in fungal genomes that are of bacterial HGT origin would effectively be identified as non-unique if the source bacterial genome(s) are available in the database, and thus would not be included in the analyses. If the bacterial genome (or progeny of the ancestral HGT donor) is not yet sequenced or available, then the identification of a near neighbor is certainly possible, given sufficient nucleotide similarity. In this case, as we mention above, we would expect punctuated read pileups on the bacterial reference genome in the region of HGT. Instead, what we generally find from this screen are multiple signatures spread among multiple regions throughout the bacterial genomes, as mentioned above and in the manuscript (lines 265-273). Such broad coverage across a bacterial genome is not consistent with what is expected from a HGT event, unless the near complete or complete bacterial genome is integrated into that of the fungal host, which has not been observed up to this point and is highly

unlikely based on our current (albeit limited) knowledge of bacterial-fungal HGT. Lastly, we would argue that bacterial HGT into fungal genomes would be indicative of bacterial associations, as an intimate association seems more likely to result in HGT than the introgression of exogenous DNA.

(viii) I believe the final analyses of the bacteria from the Illumina survey are based on OTUs, generated from ASVs. Sometimes the same OTU appears in multiple fungi. This seems surprising if the fungi were, for example, from different regions or habitats, processed by different collection dates or labs, etc. Is it possible that these differ at the ASV level? It seems unlikely that the same bacterial ASV would appear in samples from Europe vs. North America (for example) unless the 16S sequence data can't resolve them finely enough to differentiate close relatives (i.e., things are lumped somehow) or there were contamination in the processing. This can be relieved as a concern by addressing the points above.

The points concerning the definition of OTUs for this particular study have been addressed above. However, we would also point out that in terms of bacterial diversity and sequence identity within the rRNA gene, it would not be entirely surprising to find the same OTUs or ASVs in different geographic locations, due to the highly conserved nature of the 16S rRNA gene at the species (and sometimes even genus) level. So indeed, in many instances, the 16S sequence is unable to resolve distinct organisms at lower levels of taxonomy, and it can be a poor tool to identify ecotypes associated with different geographical regions.

Nonetheless, we would like to indicate that the cases the reviewer indicates correspond to a total of 102 out of 6,594 ASVs (~1.5%) were found in two collections, 32 ASVs (~0.5%) were found in three collections, and 0 ASVs were found in the four collections. Collectively, these 102 ASV sequences were classified as 53 distinct taxa, with 49 classified at the genus level (bacteria), 3 at the family level (bacteria) and the remaining one as a chloroplast. We hope that the relatively low representation in our overall dataset relieves any concerns raised by the reviewer.

We have added a statement in the results to address these results (lines 119-121): "A total of 134/6,594 of these bacterial ASVs (~2%), representing 49 distinct bacterial genera, were found in fungal isolates from two (102) or three (32) of the examined collections."

1c. Naming of taxa. The authors speak to finding many new genus-level pairings of fungi and bacteria. In doing so they rely on estimates of genus-level identification. This is problematic without further qualification for several reasons, including matching to

unnamed fungi (very common with field-collected samples such as soil fungi), obtaining multiple top matches to different taxa that are equal quality, the non-phylogenetic matching criteria, etc. Moreover, no identity thresholds or other identification criteria are listed. (i) These should all be addressed.

We added the following statements to the methods to address these concerns (lines 485-500): “The criteria for identification of fungal taxa required an Expect (E) value of 0.0 and a minimum of 95% sequence identity to an unambiguous top hit (for genus level) in UNITE. When our ITS sequences had matches to multiple closely related fungal UNITE genera with scores that passed our cutoffs (22 out of 294 isolates), the final taxonomic classification was then based on additional BLAST alignments conducted with the NCBI ITS RefSeq database, using the same classification thresholds mentioned above for the original UNITE hits. In addition, a small number of fungal isolates (10) did not meet the identity cutoffs listed, however all but one (AODJ.161.70 which had a best match at 85%) of these isolates aligned to a UNITE database reference with at least 92% sequence identity. Following published guidelines to ensure confident taxonomic classification, the ITS sequences for these 10 isolates were aligned to authenticated and/or published sequences from both the NCBI ITS RefSeq and the complete nucleotide (nt/nr) databases. Top matches were then scrutinized by comparing to other sequences with identical taxonomic classification and authenticated with closely related organisms to increase confidence. Because we cultured all isolates, growth morphology was also considered and no morphologies contradicted our classification analyses.”

(ii) I was also surprised by the large number of Ns in the ITS sequences for fungi — seems very high in our experience and it makes the sequence data suspect. Please explain and validate the sequence quality overall to provide greater confidence.

We analyzed the N content of all 294 ITS sequences (average amplicon length 592 bp). The average N content was ~2 N's per sequence. A total of 13 (out of 294) ITS sequences had 1-4% (7 to 27 N's) of their bases composed of N's, and despite this slightly lower quality, we were still able to confidently assign them (generally above 95% identity to published or curated sequences - see above and methods) at the genus level. The concerns raised by the reviewer are also, in part, the reason why we chose to be conservative in our taxonomic assignments, representing them at the genus level rather than the species level, to increase confidence in our taxonomic assignment and minimize false positives.

(iii) Sometimes previous findings are not quite represented correctly. For example the authors state that *Penicillium* was not previously known to associate with bacteria, but

wasn't such an association shown in the case of (for example) an endophyte in the *Penicillium* group by Hoffman and Arnold 2010? Also, please see: GhodsSalavi B, Svenningsen NB, Hao X et al. A novel baiting microcosm approach used to identify the bacterial community associated with *Penicillium bilaii* hyphae in soil. *PLoS One* 2017;12. DOI: 10.1371/journal.pone.0187116.

Although we tried to be as extensive as possible, we unfortunately may have overlooked elements of the literature given the large number of species that we were trying to cover. We thank the reviewer for pointing these out. We have corrected this oversight in the document and the results from both manuscripts mentioned by the reviewer are also reflected in Supplementary Table 2. We have also added a paragraph at the end of the Methods section describing the purpose and limitations of our collated list of fungal-bacterial associates in Supplementary Table 2, which we will continue to develop and plan to make available to the public as a curated, searchable resource. In addition, we have tried to be as transparent as possible with our phrasing to indicate our lack of certainty in specific cases.

We have added a section to the end of the Methods that describes our survey for previously described associations and the limitations of the survey (lines 696-709): "Literature searches were performed using relevant keywords (bacteria, fungi, bacterial associations, endohyphal bacteria, endofungal bacteria) to identify research and review articles containing previous descriptions of bacterial-fungal associations. Artificial associations, such as forced associations between bacteria and fungi not isolated from the same environment or associations involving genetically engineered strains, were excluded, as well as co-occurrence studies involving multiple fungi. Our findings are summarized in Supplementary Table 2 and include links to the references where each association is described. Due to a number of challenges such as inconsistent usage of keywords in these publications, descriptions of associations at varying taxonomic levels and descriptions of associations appearing exclusively in figures, tables and/or supplementary information, a complete summary of previous descriptions would require a separate, independent effort. Nevertheless, the data compiled in Supplementary Table 2 represents to our knowledge the first attempt to compile a comprehensive series of previously published descriptions of associations between bacteria and fungi."

1d. Transparency. I must have missed it, but could not find evidence that DNA sequences generated in this study were deposited in a publicly accessible database (the ITS sequences are in a supplement; what about the bacterial data). (i) Could accession numbers for the short-read archive or other deposition be included?

The following statement has been added to the manuscript in the Data Availability section: The ITS sequences used for classification of the fungal isolates have been deposited at NCBI GenBank and the unprocessed 16S amplicon sequencing data obtained from these isolates have been deposited in the NCBI SRA database (BioProject accession number: PRJNA738181).

(ii) Please state that all JGI genome used were fully public and not under any restrictions for use, per <https://jgi.doe.gov/user-programs/pmo-overview/policies/#data-util> (note, e.g., pre 2018 proviso).

That is correct. We now state this explicitly (lines 558-559): “To ensure that all fungal genome sequencing projects used were not under any restrictions for use, only datasets present in the public NCBI SRA database were used.”

(iii) It would be helpful to understand more about the metadata of the focal fungi, per (2), below.

2. Limited inference given the rich metadata associated with focal fungi

As written the paper entirely neglects hypotheses or predictions as a conceptual framework. While I am a fan of ‘discovery’ papers it seems that the authors could strengthen this particular work with attention to hypotheses/predictions that guide the analyses. Specifically, (i) I believe the authors must have gone into the work with some predictions based on their scholarship and their selection of focal strains. It would relieve the concern about contamination, for example, if the authors found that the geographic origin of fungi, processed in the same lab, nonetheless was key in structuring associated bacterial communities. The work could move from more of a ‘fishing and phenomenological paper’ to something more insightful, given the wealth of data the authors have generated. This relates to 3, below. I’m definitely not suggesting ‘straw man’ hypotheses, but structure to the thinking presented here could greatly strengthen the inference.

Actually, our primary objective was indeed quite exploratory - to gain a more comprehensive view (or as comprehensive as we could begin to explore) of the potential diversity and commonality of bacteria found in close association with fungi. We did this by examining data from a large number of diverse fungal isolates in *existing* culture collections, using complementary methods, and screened as many isolates as we could, regardless of where they were isolated from, their putative function or role, or even their presumptive taxonomy (in all cases this was not known prior to our work and we had to perform our own ITS analysis to obtain this information). These represent

general fungal collections that were not specifically collected or maintained or selected for our bacterial-associate screens, and often, for which we do not have ideal metadata information (including sampling strategy or experimental design). In fact, the isolation date, location, and any environmental measurements that would be useful in establishing hypotheses were unavailable in many cases.

Because our aim was to sample as broadly as possible, we were encouraged to find that many of the fungal isolates examined are from taxonomic groups without previous descriptions of bacterial associates. However, due to the limitations of our own collections and the focus of the JGI collection on taxonomic diversity (as opposed to functional class or ecological role/relevance), we proceeded with our exploratory focus without any pre-formulated hypotheses or predictions. Nonetheless, this discovery-focused project has led to hypothesis-based lines of exploration that we are currently performing, using some novel models of bacterial-fungal partners, as keenly suggested by the reviewer. Our hope is that other researchers may take advantage of our findings to posit more hypothesis driven questions and research.

We have also included a figure in a response above (1.a.v) which demonstrate fungal isolates submitted to the JGI by the same laboratory or group, despite being sequenced at a similar time, have distinct bacterial profiles, indicating that the processing lab and sequencing center do not have at the very least a consistent effect on bacterial composition.

The following statement was added to the Discussion section (lines 424-436): "We had anticipated that we may discover some potential relationships among bacterial and fungal evolutionary lineages. Initial observations within only 13 *Mortierella* are promising, with at least two different patterns emerging consisting of distinct sets of bacterial taxa (Supplementary Figure 4). It is thus tempting to speculate that some fungal taxa may harbor one of several 'core' bacteriomes, perhaps dependent on environmental pressures or its ecological niche. However broader patterns of co-occurring bacteria within related fungi were not apparent. Specific bacterial-fungal associations at the genus level appear more common than generalist associations, suggesting each fungal host harbors a unique bacteriome composed of multiple bacterial associations and that some of these associates are either transient or opportunistic. The absence of any overarching patterns of association may have been impacted by our use of diverse culture collections that utilized different culturing methods, and future explorations into the diverse nature of fungal bacterial associates will need to tailor methods so as not to impact the natural communities associated with fungi."

3. Concerns about figures and related data presentation. Fig. 1, it's up to the authors of course, but I wonder if this is a key finding of the paper — given the methodological points above and the general limitations of Venn diagrams. I struggled to see the details in Fig. 2 (so small).

While we agree there are limitations with the Venn diagrams, it is one of the few alternatives to a complex (and unreadable) heatmap that allows us to convey the most pertinent information (fungal diversity, bacterial diversity and number/diversity of associations) in a manner that allows you to compare both between our two screens, and what was known previously. A higher resolution version of Figure 2 is now included, but again the primary goal of this figure is to provide perspective on the diversity of both the fungal hosts we examined in this work and the potential bacterial associates we detected, while allowing the reader to compare that diversity to what was known prior to our work. Given the scale of the study (with hundreds of fungal isolates, each with tens to hundreds of potential bacterial associates), typical representations showing all the data, such as heatmaps, can only be included as supplementary figures due to their size. We have included a new heatmap representation of the association data collected in both screens (Supplementary Figure 1). The size of this heatmap does make it difficult to interpret, as the figure can only be examined holistically if zoomed out to the point where distinguishing different bacterial and fungal genera is challenging. Therefore, we have also included a new circular figure (Figure 3) arranged based on fungal taxonomic lineages, which demonstrates both the overall fungal diversity, including which screen each fungal genus was examined in (16S-CC, BSS, or both) and which fungal genera had no detected bacterial associates, as well as several outer rings that show the relative abundance of the most common bacterial classes detected in each fungal genus. We believe the addition of these figures address as much as possible the concerns raised by the reviewer, and highlights the key results of this work.

- Examined in screen of fungal genome sequencing projects
- Examined in culture collection screen
- Examined in both screens

B: Blastocladiomycota
C: Chytridiomycota
M: Mucoromycota
Z: Zoopagomycota

4. Other matters

Why are these archaea listed as bacteria in Fig. 3? Crenarchaeota, Euryarchaeota

Thank you for pointing out this error - we have corrected the figure and removed the archeal lineages.

Please check the spelling of fungal names — Phyllosticta is not correct (Phyllosticta). I am surprised the fungal culture collections included Lobaria, which is a lichen-forming fungus.

Thank you again for pointing this out - the spelling error has been corrected. As for *Lobaria*, it was listed within and examined as part of the JGI Mycocosm screen and not a part of the culture collection screen (Supplementary Table 5 & 6).

Sometimes the Results section included Discussion-type material (“These overlaps in the niche and trophic mode of the fungal hosts could be one possible explanation for the phylogenetic similarity found between their bacterial associates. These results collectively indicate that diverse fungi can harbor phylogenetically-related bacteria, which suggests a functional basis for these interactions.”)

In several cases, we provided some information and preliminary interpretation to provide context to the results and their relevance to the overall studies, particularly given the exploratory nature and scale of this work.

Reviewer #2 (Remarks to the Author):

The research submitted by Robinson and collaborators focus on the exploration of bacterial associates of fungi. To gain a broader overview of these fungal-bacterial associations, authors performed two different strategies:

1. They used amplicon sequencing of the 16S rDNA gene from total DNA of 294 fungal strains preserved in four fungal collections (Houston, Los Alamos, São Paulo and Neuchâtel).
2. They searched for bacterial signatures in 408 fungal genome projects available at the Mycocosm database.

Some of the newly discovered and known fungal-bacterial interactions (from 1) were examined in more detailed using microscopy (FISH or HCR-FISH).

Authors main findings are that potential bacterial associates of fungi are more widespread and diverse than anticipated, but no clear patterns were observed in terms of the number of bacterial associates and/or the bacterial lineages associated with

fungus hosts (similar to findings by Hoffmann & Arnold AEM 2010). Altogether, the results presented here support the idea that fungal biology, and potentially the biology of plants and animals that harbor also a mycobiota, may be affected by these associations.

Despite the study is of interest and presents a broad investigation, the major results are, in my opinion, not concisely and well presented (in an informative way). Thus, the conclusions lack support and clear evidence.

Authors, although provided with ITS fungal sequences or (nearly) full genome data, do not present in a clear way the fungal taxa included in the study (a phylogeny) and the sample size of each group (Phylum, family, genus) for each of the two approaches. Authors could follow to some extent the way Myers et al., mBio 2020 presented the fungal taxa screened and the “potential bacterial (instead of viral)” associates found. For the reader it is difficult now to really comprehend and see (in figures!) which fungal groups were examined and which bacterial associates were found (the likely best representation of this was Supplementary Figure 6, in my opinion).

Our original intent was indeed to have a phylogenetic representation next to a heatmap that displayed the bacterial diversity found per isolate, but this was thwarted by the sheer diversity of bacterial taxa discovered, which was difficult to visualize in a regular manuscript figure due to the space required (though we have now included such a representation in supplementary information, Supplementary Figure 1). Our effort to demonstrate this intractable visualization issue was in fact presented as an overview in Figure 6b, where the complexity of relationships detected in both screens is represented visually. While a simpler tree-based view such as the one depicted by Myers et al. could be presented, we did not feel it would convey the breadth of relationships discovered (particularly the breadth of bacterial diversity). Furthermore, we did not wish to focus attention on the fungal diversity alone. We have attempted to display such relationships in a new high-level figure (Figure 3), which demonstrates both the overall fungal diversity, including which screen each fungal genus was examined in (16S-CC, BSS, or both) and which fungal genera had no detected bacterial associates, as well as several outer rings representing the relative abundance of the most common bacterial classes detected in each fungal genus. We welcome any additional thoughts the reviewer has with respect to figures, but believe this latest figure addresses the bulk of the reviewer's main concerns.

In addition, we have provided tables delineating every detected association for both screens (Supplementary Table 3 & 6) with complete taxonomic breakdowns for both the

bacterial and fungal partners, which due to their segregated format, should also facilitate queries about sample size and diversity in the context of detected associations.

Also, sequence data (fungal ITS sequences and 16S barcoding) seems not to be deposited in public databases, as no accession numbers and/or bioprojects are referenced in the main text. Please include.

The following statement has been added to the manuscript in the Data Availability section: “The ITS sequences used for classification of the fungal isolates have been deposited at NCBI GenBank and the unprocessed 16S amplicon sequencing data obtained from these isolates have been deposited in the NCBI SRA database (BioProject accession number: PRJNA738181).”

I have some concerns about the methodology used for the 16S-CC screen. It is unfortunate that protocols were, to some extent, differently performed by the four participating Labs. Some used chloramphenicol for the subculturing of the fungi, some did not. Chloramphenicol is a broad spectrum antibiotic, so it definitely can affect the “bacteriome” associated with a fungus. This has been already reported in *Mucor hiemalis* (Schulz-Bohm et al., Fungal Genet Biol. 2017). Did this treatment affect the number and taxa of the potential bacterial associates found? Authors should quantitatively account for this aspect. Have chloramphenicol-treated fungi on average less bacterial OTUs than non-treated? Have they different bacterial taxa when compared to other fungi of the same genera that were left untreated?

As we mentioned in a response to reviewer 1 above, our goal was to screen as many fungal isolates available to us at the time. While we acknowledge that the different protocols (including use of chloramphenicol in some cases) likely skewed or at minimum, reduced the diversity of the bacteriome in those fungal samples, we would argue that any/all lab-specific culturing condition(s) would also impact the bacterial profile (including time in culture and conditions such as temperature, humidity, sub-culturing techniques and media used).

We did indeed observe an impact on overall alpha diversity for the culture collections that had chloramphenicol-treated fungi (fewer bacterial taxa/OTUs per isolate than non-treated). Below is a box-whisker plot showing this impact. Given that no isolates were exposed to both treatments (chloramphenicol and no chloramphenicol), we limited the comparison shown in the boxplot to fungal genera found in both collections that underwent antibiotic treatment (Houston and Brazil) and those that did not (LANL and Neuchatel). This resulted in us examining 106 fungal isolates (66 treated and 40 un-

treated) representing 7 fungal genera (*Aspergillus*, *Cladosporium*, *Clonostachys*, *Fusarium*, *Penicillium*, *Purpureocillium*, *Trichoderma*).

These specific results were not called out in the manuscript, as despite the reduced number of bacterial associates in the treated samples, there were still bacterial signatures observed.

Regarding the bacterial associate community composition and differences among treated vs untreated fungi - this question cannot be answered with the isolates examined, since all treated isolates were from one collection while untreated isolates of the same genera were from other culture collections, thus obfuscating the impact of the antibiotic and other aspects of the collection (including source of the isolates and local culturing conditions, etc.). A dedicated study, such as the *Mucor hiemalis* study mentioned by the reviewer, could potentially address this question.

The following statement was also added to the Discussion section to indicate the potential impacts of our diverse culturing methods (lines 432-436): “The absence of any overarching patterns of association may have been impacted by our use of diverse culture collections that utilized different culturing methods, and future explorations into

the diverse nature of fungal bacterial associates will need to tailor methods so as not to impact the natural communities associated with fungi.”

By reading the methods, I was unable to clearly identify if authors have biological replicas of the 16S barcoding of some (or most) of their fungal samples. For the microbiome of plants and animals, 3 biological replicas are common. Do authors have biological replicas? were biological replicas similar among them?

No biological replicas were used for this study. We utilized methods similar to seminal studies, such as those by Hoffman and Arnold (Diverse Bacteria Inhabit Living Hyphae of Phylogenetically Diverse Fungal Endophytes). Similar to this study, our primary interests were to describe the overall diversity rather than conduct comparisons across isolates, or collections (or the other generally limited metadata) for these isolates. Given the large number of fungal isolates examined using this method, biological replicates would have put our total sample number near 900, which was not a possibility given the scope of this project as we wished to screen as many isolates as possible. We understand the use of replicates in larger eukaryotes with well-defined compartments and organs, allowing for biological replication. Certainly, there have been several studies probing the large differences for example between microbial communities present within a single human mouth, when sampling in different areas. There are also many observations of temporal dynamics from longitudinal studies of gut microbiomes. In our case, most of the fungal isolates have been maintained in culture for years and we utilized a substantial amount of fungal mycelia for sequencing, which could indicate that any differences observed in biological replicates may be the result of spatiotemporal dynamics of the fungal bacteriome. Such spatiotemporal dynamics have been observed in fungi, but like observations in larger eukaryotes, they are not well understood and would require a separate dedicated study to investigate adequately.

were negative controls also included (known bacteria-free fungi such as some reported strains of *R. microsporus*)?

We did not sample any ‘known’ bacteria-free fungi as control, but, as indicated above in the answers to Reviewer 1, extraction/no-template controls were included for each collection and extraction kit. Any bacterial ASVs with identical taxonomic classifications to those detected in these control samples were excluded from the presented results (i.e. if a single *Bacillus* ASV was found in any of the control samples for a particular collection, all *Bacillus* ASVs were excluded from that collection). This information is in the Methods section.

The barcoding data allows not only to identify which taxa is present, but also its relative abundance. I wonder why authors do not present this information. The analysis of this data could be further used to infer which symbionts are more abundant in a fungal host, and which ones could be secondary symbionts. This piece of information could be later validated using FISH with specific probes. Moreover, authors could also use some tools of microbial ecology (NMDS, Permanova/AnoSim, etc.) to further support their findings, especially to test if the fungal host and its lineage (or the use of chloramphenicol and/or the ecological niche) is a factor influencing bacterial community assembly.

The reviewer brings up many interesting questions regarding the nature of these interactions, and we are indeed interested in such questions. That these questions can now be asked is an indicator of the relevance of this preliminary exploration into fungal bacterial associates and is an example of what hypotheses may now be further explored in a more directed fashion.

In addition to the spatio-temporal concerns mentioned above (that could invalidate assumptions regarding relative abundance ratios), it is important to consider we utilized nested amplification, which may skew the relative abundances and complicate using this data quantitatively. While we are still unclear of the value of focusing on relative abundances, we have, however, included the complete ASV table generated by QIIME2 for the 16S-CC screen, which reports the relative abundance of all detected ASVs (Supplementary Table 4).

The following statement has been added to the Discussion section (lines 424-436) to address the questions and concerns raised by the reviewer: “We had anticipated that we may discover some potential relationships among bacterial and fungal evolutionary lineages. Initial observations within only 13 *Mortierella* are promising, with at least two different patterns emerging consisting of distinct sets of bacterial taxa (Supplementary Figure 4). It is thus tempting to speculate that some fungal taxa may harbor one of several ‘core’ bacteriomes, perhaps dependent on environmental pressures or its ecological niche. However broader patterns of co-occurring bacteria within related fungi were not apparent. Specific bacterial-fungal associations at the genus level appear more common than generalist associations, suggesting each fungal host harbors a unique bacteriome composed of multiple bacterial associations and that some of these associates are either transient or opportunistic. The absence of any overarching patterns of association may have been impacted by our use of diverse culture collections that utilized different culturing methods, and future explorations into the diverse nature of fungal bacterial associates will need to tailor methods so as not to impact the natural communities associated with fungi.”

The tools suggested by the reviewer (NMDS/Permanova) typically take advantage of well-defined metadata, which often revolves around treatments or conditions (or environmental measurements) selected or imposed to address a specific biological question. In the case of the isolates used in this study, the metadata is very limited, requiring such investigations to focus on comparisons between culture collections or antibiotic treatment, and even then, our experimental design does not allow for direct comparisons between these conditions (i.e., how does antibiotic treatment affect the same fungal isolate or how does being cultured and maintained in another lab affect the fungal microbiome). This is because different isolates in the different collections were subjected to different treatments. Furthermore, these statistically driven methods would be less impactful without biological replicates to control for variance within each isolate. Currently, our team is developing more complex community analysis tools that take all these factors into consideration to determine any additional factors that could explain the composition of fungal microbiomes such as predicted functional capability of bacterial associates or inferred functional fungal guild, rather than just their taxonomy.

FISH and HCR-FISH were only used in the context of the study to identify bacterial cells in fungi, but not to show any example of a complex/diverse bacterial community within a host. Authors report that some of the screened fungi have up to 100 OTUs and an average of 34 per strain (lines 122-3). Unfortunately, these numbers can not be directly evidenced from their microscopic studies.

The use of taxonomic specific probes would be required to distinguish individual bacteria and to visually observe diversity in validations using for instance FISH. It is important to recall that we only obtained a part of the 16S rDNA region in our culture collection screens, which limits our ability to design probes specific for each bacterial taxa. In addition, the process of imaging diverse fungi using FISH is not trivial even with universal probes, since substantial effort is required to adapt FISH techniques to each and every new fungal isolate examined. While the type of microscopy suggested by the reviewer is highly desired, we believe such an effort would in itself be deserving of its own manuscript. Our group has successfully developed and tested a few isolate-specific probe sets, and we have added an example of these results and the relevant methods to the manuscript (Figure 5). This newly included figure demonstrates the presence of *Lacunisphaera*, a previously undescribed bacterial associate of fungi, in an isolate of *Aspergillus*, a fungal genus which to our knowledge does not have previous descriptions of bacterial associates.

Based on the micrographs displayed in Figure 3, I was unable to observe bacterial cells in the outer surface of fungal cells (epibionts). Authors claim that their study incorporates both external and internal associates of fungi, but I am afraid that the methods used do not allow them to separate these two groups.

Distinguishing internalization vs externalization requires, at minimal, confocal microscopy, which is challenging when using diverse fungi as mentioned above, and even then, there is still a degree of uncertainty in some cases. We indicated that our study encompasses investigations of potentially external and internalized bacterial associates, as our methods neither exclude nor target either one specifically. Therefore, the results and discussion focus on the diversity of potential associates, rather than their internal/external characterization.

As an aside, we have also included a new supplementary movie which demonstrates the results of utilizing confocal microscopy with probes designed to target a specific bacterial genus. In doing so, it appears that this particular bacterial associate is internalized within the fungal hyphae (Supplementary Movie 1). However, we want to re-emphasize that while spatial characterization was examined and presented in this particular result, it was never a primary focus of our work to distinguish between potential internal and external bacterial associates, largely due to our primary goal of expanding knowledge on the potential diversity of associations between fungi and bacteria, regardless of spatial characterization. It remains an area of great interest however for future studies.

It would have been also desirable that fungal cultures used for DNA extraction were derived from single spores (monosporic cultures) to make sure associations are tight, and not only circumstantial.

Starting cultures from spores can be complicated, as conidiation (asexual spores) either does not occur or occurs at low levels in some isolates and the production of sexual spores can require compatible mates (heterothallic) or specific growth conditions (homothallic). Given our limited knowledge on the fungal microbiome, it may also be possible that spores have distinct microbial associates since they are distinct structures (as observed with members of the Glomeromycota and their *Candidatus Glomeribacter gigasporarum* partners) or that establishment of the fungal bacteriome occurs after germination from the environment similar to what is thought to happen with animal gut microbiomes and most of the components of the plant microbiome.

The finding that all strains screened in the 16S-CC were associated with at least one bacterial associate sounds suspicious to me. In previous published screenings the frequency of finding bacterial symbionts in several fungal hosts has never been 100% (see please Hoffmann & Arnold AEM 2010, Desiró et al. ISME J 2014, Toomer et al. Mol Ecol 2015, Dolatabadi et al. Fungal Biol 2016, Okrasińska et al. AEM 2021, etc.). Thus, it is very likely that authors are reporting spurious associations.

Detecting at least a potential bacterial associate in every fungal isolate was also somewhat surprising to our team, but not completely unexpected given our approach, which is unique in many respects as compared to the studies mentioned by the reviewer. There are a number of differences between the goals and the methods used in the previously published screenings that reduce the frequency of detecting bacterial associates relative to our employed methods. For example, two of these studies use surface sterilized spores (Desiro et al. and Toomer et al.), coupled with specific primers that target expected (and specific) associates of interest, both of which are aimed at limiting the diversity of bacterial associates compared to our investigations using general primers and fungal mycelia. Another study only examined 16S from bacterial colonies isolated from disturbed mycelia (Bolatabadi et al.) imposing growth on specific media as a selection criteria prior to delineating the bacterial taxa. These studies also all utilize cloning and Sanger sequencing, a method that is vastly less sensitive in detecting taxa present at low abundance, than our high-throughput Illumina based approach. While these methodological differences are certainly one explanation for the increased prevalence of bacterial associates in our examined fungal isolates, we have also exerted diligence, both in the responses above and throughout the manuscript, to indicate all detected and presented associations are putative.

The following statement was added to the Discussion section to facilitate comparisons between our results and similar previously published studies (lines 407-415): “In our 16S amplicon screen of diverse fungi belonging to multiple culture collections, at least a single putative bacterial associate was detected in every examined isolate. This detection rate was unexpected given that results from previously published surveys usually contain several isolates lacking any potential bacterial associates. However, differences in the applied methodology of these previous surveys such as the use of primers targeting specific taxonomic groups or the treatment of fungal tissue to eliminate or reduce external bacterial associates, compared with the more sensitive methods applied here, could explain the increased discovery rate, as our methods are designed to generally detect any potential bacterial associates.”

With respect to the genome-based strategy (or BSS), one of the key aspects missing is, in my opinion, to show a quantitative overview of the bacterial signatures found. From reading the actual manuscript, it is hard to visualize:

1. From each fungal genome analyzed, how many sequences (contigs and their lengths) were putatively associated with bacteria? Which proportion do they represent from the total DNA? What is the relative abundance of each bacterial group associated with a fungal host?...

All classification for the BSS screen was done at the read level and these were classified based solely on hits to portions of the bacterial genomes that are unique to a genus. Supplementary Table 7 shows the summary results from the BSS screen, including the read count (i.e., total number of ‘hits’ per genome), total base-pairs mapped, number of mismatches, and the linear coverage and depth of coverage of the reference genome in question. To provide a quantitative overview of the bacterial signatures found within these fungal sequencing projects, we have provided the figure below as Supplementary Figure 7. The large number of sequencing projects examined and the high variance in both total sequencing performed per fungal project, and number of bacterial reads identified makes insightful visual representations challenging, which is why we opted to represent this data in a table, which can easily be queried to examine relationships between recorded variables.

The number of reads associated with any bacterial reference genome is impacted by the relative abundance of the bacterial cells in the sample, including relative to the number of fungal cells (note, there could be spatial heterogeneity in the distribution of any putative bacterial associate), and the size and copy number of the bacterial genome per cell, and of the fungal genome as well. Because most projects have a very small proportion of non-fungal data and fungal genomes themselves vary greatly in sequencing coverage and total amount of sequencing, an ideal scenario would be to have low abundance spike-in controls in each genome project to see if we could estimate true relative abundance based on a standard curve. Unfortunately, this is not the case, thus we only have relative abundances which we can use to calculate relative to the host genome (where we can normalize for genome size or depth of coverage) or relative among the bacterial community members.

In this approach, I liked very much that authors investigated SNPs in the bacterial signatures associated with *B. cereus*, *E. coli* and *A. johnsonii* from several fungal hosts. These results do show that similar bacterial symbionts are putatively associated with diverse fungal taxa, and that these bacterial strains are probably similar to bacteria that occupy other ecological niches. In Figure 5, it would be desirable that authors add an icon to specify the fungal phylum of occurrence and/or the ecological niche of the bacterium.

While our knowledge of ecological niche is quite restricted for some of the fungal isolates examined, we agree that the figure could benefit from an indication of the fungal

phylum for each isolate, which we have added to Figure 7 (previously Figure 5). Many of the bacterial genomes used for these phylogenetic comparisons also lack sufficient metadata related to their ecological niches or functions, thus we cannot indicate ecological information in the figure.

I strongly suggest authors to think in ways in which both approaches (16S-CC and BSS) could render similar, informative and concise figures that allow readers to easily make comparisons and reach conclusions.

Most actual figures are either too simple (figure 1) or too complex (figure 2, 4) to deliver valuable information.

We have had many discussions on how best to represent these complex data in figures that are as informative as possible. It is indeed difficult to provide such rich and nuanced data concisely, and selected figures that show some of the specifics outlined in the text. We have included additional figures that we hope aids readers in comparing not only methodological approaches to detecting bacterial associates in fungi, but also to examine the fungal taxa and associated bacterial diversity. For example, the newly included Figure 3 allows readers to compare both the diversity of fungal associates examined in both screens and the profiles of the most prevalent bacterial classes detected across both screens for each examined fungal genus.

I also recommend authors to discuss in more detail the limitations of their study, and to provide the most promising perspectives gained through the analyses done. Are there

specific fungal genera that would enable a broad understanding of BFI? Or studies should focus more on the same ecological niche? are 35 bacterial OTUs stably associated with “axenic” fungi?

Current knowledge of BFI indicates they are quite taxonomically broad (both fungal and bacterial), and while little is known about their impacts on biological or ecological function, we predict these will be equally diverse. The questions asked by the reviewer are all equally valid and again, help to highlight the types of questions that immediately arise given our findings. While we do intend to examine some of these questions more specifically, we generally believe, at least for the first question, that no specific fungal genera could provide a sufficiently broad understanding of BFI to encompass the diversity of interactions observed. This study is intended to be a starting point, indicating really for the first time, that BFI are more common than previously anticipated, and we believe this work will encourage investigations into the impacts of BFI in specific taxa or ecological niches. Stability of any associations would require an independent study, as we expect it to differ among both bacterial and fungal taxa, and be influenced by factors such as culture and environmental conditions.

In the Discussion section we also include a statement regarding the perspectives gained through this work (lines 438-447): “While this work provides an important overview and perspective on the diversity of bacterial-fungal associations and the potential complexity of the fungal bacteriome, the underlying mechanisms responsible for these associations remain largely unknown. Elucidating overarching mechanisms responsible for establishing and maintaining bacterial-fungal associations has in the past been hindered in part by the limited diversity and number of described associations, an obstacle addressed directly in this study. Detailed examination of some of these diverse bacterial-fungal partner pairs will help elucidate key genes and pathways that govern bacterial-fungal interactions. Increased knowledge of these underlying mechanisms will be paramount to help predict the biological outcomes of these associations under changing environmental conditions, and their potential impact on ecosystem functioning.”

Reviewer #3 (Remarks to the Author):

The manuscript by Robinson and colleagues reports results of two surveys targeting bacteria associated with fungi: (1) empirical investigation of 294 fungal isolates from four culture collections (16S-CC screen) and (2) computational examination of 408 fungal genomes sequenced by JGI (BSS). While both efforts yielded evidence of numerous bacterial taxa associated with focal fungi, which is a source of valuable

information, presentation and interpretation of these data requires adjustments, as discussed below:

1. The study is a “fishing expedition” devoid of any ecological context. Therefore, its findings are at best observations that could be used to formulate hypotheses for further studies. In particular, finding DNA sequences of *Bacillus cereus*, *E. coli* and *Acinetobacter johnsonii* in multiple fungal genomes should not be over-interpreted. The prudent hypotheses here would be that these bacteria are indeed associated with their respective hosts. If these hypotheses are supported by empirical tests of living fungal cultures, then patterns of associations could be examined.

We agree entirely that the goal of this study was to examine more broadly than any prior study, the putative diversity and prevalence of bacterial associates among diverse fungi. As stated above, we also indicate that one of the goals of this study was to provide a foundation to facilitate more directed and hypothesis-driven investigations in the future. We did make an effort to balance our general observational findings with potential interpretations suggested by the results. For example, in the manuscript, we state for the *B. cereus* examinations (lines 366-368): “These results indicate that diverse fungi may harbor phylogenetically related bacteria, and that multiple lineages within a bacterial species may be able to form such cross-kingdom associations.”, which limits over-interpretation of the results as it does not state certainty about the associations.

A separate reviewer above commented that they valued the comparison because it suggested the fungal associating bacterial lineages were distinct from both one another and other closely related lineages that have not been described as fungal associates and occupy other ecological niches, as it builds the foundation for more hypothesis driven investigations.

2. I have a problem with framing of the rationale for this study (lines 53-57), suggesting that community-level ecological studies of fungal-bacterial interactions are not informative. Most of the citations used to support this notion are review papers because community-level field studies are in their infancy and these ecological data are still limited. However, once such studies start yielding data, they will be vastly superior to the approach taken by the authors.

We believe the reviewer has misinterpreted our intention within lines 53-57. It was meant to highlight the most relevant and most closely aligned work with our question of BFI diversity, and to highlight the knowledge gap that remains and that our study intends to fill. We wished to point out that because such microbial community ecological approaches do not address our primary question regarding bacterial associates of

diverse fungi, that we required a different approach. The framing of our rationale was to indicate that while community-level studies of complex environments are important and have revealed interesting results, our current limited knowledge of bacterial-fungal interactions makes it difficult to confidently distinguish between ‘associations’ and co-occurrence (which does not necessarily indicate an ‘association’). For example, there could be bacteria and fungi that are both adapted and thrive under similar environmental conditions, but never actually interact with each other, which would be highlighted in a co-occurrence study, but likely be excluded when examining for BFI specifically. We have nonetheless modified the paragraph to clarify our rationale and justification (lines 50-53).

3. Line 97. Please list these collections here and describe briefly how sequence data were generated. With M&Ms in the back, it is difficult to follow the Results.

This dataset was intended to encompass the broadest range of fungal diversity possible and the collections, while distinct, represent general fungal isolate collections from four institutions. We have added a statement briefly describing how the sequencing data was produced and taxonomically classified, and indicated more detailed information is provided in the Methods for anyone who is curious.

4. Line 206 and 400. The authors keep referring to fungal hyphae/mycelium as tissues. Please correct.

We have altered the phrasing to hyphae/mycelium/individual where appropriate.

5. Lines 283-288. Were Mycetohabitans, Mycoavidus and Mycoplasma-related bacteria not detected?

We recognize that these bacteria are frequent associates of diverse Mucoromycota, however, they are not necessarily present in every fungal isolate in this phylum and had not thought it necessary to point this out explicitly. For example, BRE was only found in 37% (22 out of 59) species of *Mortierella* in the study by Takashima et al. 2018. Given our examination of only 28 Mucoromycete isolates, we knew it was possible that bacteria from these lineages may be present at low frequencies or absent all together.

ASVs classified as *Mycoavidus* were in fact detected in some of the examined *Mortierella* isolates from the LANL collection, but identical ASVs were found in 1 out of 10 control samples for that collection, requiring us to exclude them from the results, given our strict quality control measures.

Representatives of *Mycetohabitans* were not found in any of our examined isolates, but several bacterial associates from the BSS screen were classified as *Mycoplasma*. Supplementary Table 3 contains information regarding all associations in a format that is easy to query based on fungal or bacterial diversity.

6. I was intrigued by the networks of fungal-bacterial associations presented in Figure 4. However, upon closer inspection, I realized that only panels a and c are informative; panel b is vastly confusing and needs to be explained better. Perhaps redrawing this network as two separate networks: 16S-CC and BSS would work better. It would be also useful to know what tool was used to draw this network.

Panel b in Figure 4 (now Figure 6) was included, and we believe important, to demonstrate the level of complexity of the overall network of potential associations detected across our screens at the genus level. The data for that network figure was visualized using Cytoscape, which we now indicate in the figure caption.

7. In the References, please make sure that journal titles are capitalized.

Thank you, we corrected the references formatting.

REVIEWERS' COMMENTS:

Reviewer #1 (Remarks to the Author):

I am delighted to review the revised manuscript and response to reviews of the original submission (for which I was reviewer 1).

I would like to compliment the authors on their careful and thorough response to reviews. I know how much energy and time that takes, and the authors have done it well. While I do not agree with all points they have made, I feel they've done an admirable job of addressing my questions and suggestions, increasing the transparency of the manuscript, increasing confidence in their results, and improving the submission as a whole. I also feel that the valid points raised by the other reviewers have received appropriate attention. To the authors, well done, and thanks.

I have just a few points for the authors to please consider:

I recommend avoiding 'adapt' (line 51). How about 'that co-occur' and skip thrive too (organisms can occur but not thrive..., and yet still interact...).

Line 117, 'rolled up into' sounds pretty colloquial; can that be formalized and clarified?

Line 120, distinct is redundant in my view, and can be removed (by definition genera are distinct from each other).

In the introduction, the authors could strengthen their case with statements about their controls (under stringent quality control, with extraction blanks...) to reduce a barrier of skepticism in the reader. Casual readers rarely scrutinize the methods, in my experience, so a few words about the care you took in the work could strengthen the 'visible' parts of the paper. I recommend similar language to be included, briefly, in the discussion. I think the effect would be to strengthen the paper.

In the same paragraph I might advocate for protist instead of protozoa, and suggest in the last section (and really, wherever the authors can), a bit more introspective a tone: for example, instead of...

This raises a multitude of questions at two scales: first, the potential role and impacts of the fungal bacteriome on the fungal host...

please consider something like this...

This raises a multitude of questions: to what extent are bacteria transient or long-lasting in their occurrence with fungi? How frequently do fungi serve as substrates for bacteria with wide ecological niches, or as hosts for specialized bacteria? Considering both transient co-occurrences and more lasting associations, what are the potential roles and impacts of the bacteriome on fungi?

More generally, I'd take a little care with 'host' and 'association' still, as illustrated in my example above.

As a comment, I really like the ASVFISH proof of concept. Cool.

Thanks for the updated methods and figures, indeed where many of the important changes have been made that greatly strengthen the paper and its inferences.

- A. Elizabeth Arnold, University of Arizona

Reviewer #2 (Remarks to the Author):

This revised version of the manuscript successfully addresses most of the concerns/suggestions made by us reviewers. Great job!

I truly think the manuscript has gained a lot in this round of revision, so I am happy I can recommend now its acceptance and publication. Congratulations to all authors!

The only minor things authors can still consider to improve are, in my opinion, the following:

1. North and South America represent only one continent, the American continent. Thus, I think authors should change line 30 of the Abstract that says “..spanning three continents..” to “.. spanning North America, South America and Europe” to be accurate.
2. Line 80. I suggest to change “nearly every fungal phylum” to “six out of the eight recognized fungal phyla”. This is more precise and leaves place for more discoveries (terra incognita)!
3. Line 144. As Actinobacteria represents a phylum and not a class, you may say “bacterial lineages” instead of “bacterial classes”. This inconsistency is repeated along the text (line 147, 960).
4. Line 918. I think you mean “The confocal microscopic studies were funded by...” or did you get a new confocal microscope for this study?.
5. Try to integrate most of the supplemental material in one single PDF that has a heading with the title of the manuscript, the authors and the index and description of all supplemental figures, tables, data and movies that are associated with the work. I could not see Supplemental Figure 1 (corrupted file?). Labels/names of txt files were also confusing. So, please correct that people can understand and find all the resources you mention in the text.

Reviewer #3 (Remarks to the Author):

The revised manuscript by Robinson and colleagues is greatly improved. I have only a few minor suggestions:

Line 29. Should be: “Both a 16S rDNA-based...”

Line 134. I would say: “This pattern suggest...”

Line 425. Are these Mortierella isolates or species?

Line 504 and 522. Should be “16S rDNA”.

Reviewers' comments:

Reviewer #1 (Remarks to the Author):

I am delighted to review the revised manuscript and response to reviews of the original submission (for which I was reviewer 1).

I would like to compliment the authors on their careful and thorough response to reviews. I know how much energy and time that takes, and the authors have done it well. While I do not agree with all points they have made, I feel they've done an admirable job of addressing my questions and suggestions, increasing the transparency of the manuscript, increasing confidence in their results, and improving the submission as a whole. I also feel that the valid points raised by the other reviewers have received appropriate attention. To the authors, well done, and thanks.

I have just a few points for the authors to please consider:

I recommend avoiding 'adapt' (line 51). How about 'that co-occur' and skip thrive too (organisms can occur but not thrive..., and yet still interact...).

We have changed the wording in this sentence to align with the suggestion of the reviewer: "For example, some bacteria and fungi that co-occur under similar environmental conditions, but never actually interact with each other, could be highlighted in a co-occurrence study, but likely be excluded when examined for BFI specifically."

Line 117, 'rolled up into' sounds pretty colloquial; can that be formalized and clarified?

The wording in this sentence has been adjusted to be made clearer and more formal: "Across all examined fungal isolates, a total of 6,594 16S amplicon sequencing variants (ASVs) were clustered into 705 bacterial operational taxonomic units (OTUs), based on ASV taxonomic classification (see Methods), representing 27 phyla, 53 classes, 108 orders, 213 families and 546 genera."

We have also updated similar descriptions in the Methods section to reflect this change in language (i.e. 'rolled up' to 'clustered').

Line 120, distinct is redundant in my view, and can be removed (by definition genera are distinct from each other).

We have removed the redundant descriptor:

"A total of 134/6,594 of these bacterial ASVs (~2%), representing 49 bacterial genera, were found in fungal isolates from two (102) or three (32) of the examined collections."

In the introduction, the authors could strengthen their case with statements about their controls (under stringent quality control, with extraction blanks...) to reduce a barrier of skepticism in the reader. Casual readers rarely scrutinize the methods, in my experience, so a few words about the care you took in the work could strengthen the 'visible' parts of the paper. I recommend similar language to be included, briefly, in the discussion. I think the effect would be to strengthen the paper.

The following statements have been added to the Introduction:

“Stringent quality control standards and procedures were employed for both screens, to increase confidence in the accuracy of the observed results.”

and Discussion:

“Quality control standards and procedures were employed throughout our investigations to increase confidence in the accuracy of the presented results and aid in interpretation.”

In the same paragraph I might advocate for protist instead of protozoa, and suggest in the last section (and really, wherever the authors can), a bit more introspective a tone: for example, instead of...

This raises a multitude of questions at two scales: first, the potential role and impacts of the fungal bacteriome on the fungal host...

please consider something like this...

This raises a multitude of questions: to what extent are bacteria transient or long-lasting in their occurrence with fungi? How frequently do fungi serve as substrates for bacteria with wide ecological niches, or as hosts for specialized bacteria? Considering both transient co-occurrences and more lasting associations, what are the potential roles and impacts of the bacteriome on fungi?

More generally, I'd take a little care with 'host' and 'association' still, as illustrated in my example above.

The final sentence of the Introduction section was edited to conform with these suggestions:

“This raises a multitude of questions: how frequently do fungi serve as hosts for specialized bacteria, or as potential substrates for bacteria with broad ecological niches? To what extent are bacteria transient or more persistent in their occurrence with fungi? Considering both transient co-occurrences and more persistent associations, what are the potential roles and impacts of the bacteriome on fungi, including impacts on interactions with other microscopic (e.g. protists) or macroscopic (e.g. plants and animals) organisms?”

Additionally, we have adjusted the language where appropriate throughout the manuscript to address the reviewer's concerns about the use of the terms 'host' and 'association'.

Reviewer #2 (Remarks to the Author):

This revised version of the manuscript successfully addresses most of the concerns/suggestions made by us reviewers. Great job!

I truly think the manuscript has gained a lot in this round of revision, so I am happy I can recommend now its acceptance and publication. Congratulations to all authors!

The only minor things authors can still consider to improve are, in my opinion, the following:

1. North and South America represent only one continent, the American continent. Thus, I think authors should change line 30 of the Abstract that says “..spanning three continents..” to “..spanning North America, South America and Europe” to be accurate.

The statement has been updated to reflect this suggestion:

“Both a 16S rDNA-based exploration of fungal isolates from four distinct culture collections spanning North America, South America and Europe, and a bioinformatic screen for bacterial-specific sequences within fungal genome sequencing projects, revealed that a surprisingly diverse array of bacterial associates are frequently found in otherwise axenic fungal cultures.”

2. Line 80. I suggest to change “nearly every fungal phylum” to “six out of the eight recognized fungal phyla”. This is more precise and leaves place for more discoveries (terra incognita)!

The statement has been updated to reflect this suggestion:

“This work provides a considerably more comprehensive exploration of the fungal bacteriome by examining over 700 fungal isolates, including 366 fungal genera (nearly ten times the amount in all previous examinations) and multiple representatives from six out of the eight recognized fungal phyla.”

3. Line 144. As Actinobacteria represents a phylum and not a class, you may say “bacterial lineages” instead of “bacterial classes”. This inconsistency is repeated along the text (line 147, 960).

Thank you for pointing this out. All statements indicated by the reviewer have been updated in a similar fashion:

Line 147 and 150: “Members of the bacterial lineages Betaproteobacteria (found in 271 of 294 fungal isolates), Gammaproteobacteria (258), Alphaproteobacteria (256), Actinobacteria (247) and Bacilli (219) were detected most frequently across all examined fungal isolates and OTUs assigned to these five lineages were also responsible for a large (6,411; ~82%) proportion of all 7,830 detected bacterial-fungal associations (Figure 3).”

Line 963: “The innermost ring indicates the screen used to detect these bacterial-fungal associations, and the outer five rings represent the top five bacterial lineages (from inner to outer ring: Alphaproteobacteria, Betaproteobacteria, Gammaproteobacteria, Actinobacteria and Bacilli) most frequently detected in our screens.”

4. Line 918. I think you mean “The confocal microscopic studies were funded by...” or did you get a new confocal microscope for this study?.

Thank you, however after discussing this point, we have decided to exclude the statement from the Acknowledgements section.

5. Try to integrate most of the supplemental material in one single PDF that has a heading with the title of the manuscript, the authors and the index and description of all supplemental figures, tables, data and movies that are associated with the work. I could not see Supplemental Figure 1 (corrupted file?). Labels/names of txt files were also confusing. So, please correct that people can understand and find all the resources you mention in the text.

The supplemental figures and text have been combined into a single PDF. Supplemental Figure 1 is a very large heatmap, and given that this issue was not mentioned by the other reviewers, we anticipate the file is not corrupted, but rather had issues rendering based on the method being used to view the file by this particular reviewer. Due to the scale and size of this heatmap, we have suggested that this figure be provided as a separate file in the editorial requests file.

Reviewer #3 (Remarks to the Author):

The revised manuscript by Robinson and colleagues is greatly improved. I have only a few minor suggestions:

Line 29. Should be: “Both a 16S rDNA-based...”

We have adjusted this statement:

“Both a 16S rDNA-based exploration of fungal isolates from four distinct culture collections spanning North America, South America and Europe, and a bioinformatic screen for bacterial-specific sequences within fungal genome sequencing projects, revealed that a surprisingly diverse array of bacterial associates are frequently found in otherwise axenic fungal cultures.”

Line 134. I would say: “This pattern suggest...”

We have adjusted this statement:

“This pattern suggests that specific, or possibly opportunistic, interactions are also not uncommon”

Line 425. Are these Mortierella isolates or species?

This statement was referring to isolates, and has been updated to be more explicit:

“Initial observations within only 13 Mortierella isolates are promising, with at least two different patterns emerging consisting of distinct sets of bacterial taxa (Supplementary Figure 4).”

Line 504 and 522. Should be “16S rDNA”.

Both titles have been updated to reflect this suggestion:

Line 507: **Amplification of 16S rDNA from fungal isolates**

Line 525: **Analysis of 16S rDNA amplicons obtained from fungal isolates**